# Robust models of disease heterogeneity and control, with application to the SARS-CoV-2 epidemic

**Kory D. Johnson**[1]*, **Annemarie Grass**[2], **Daniel Toneian**[2], **Mathias Beiglböck**[2], **Jitka Polechová**[2]

**1** Institute of Statistics and Mathematical Methods in Economics, TU Wien, Vienna, Austria, **2** Department of Mathematics, University of Vienna, Vienna, Austria

☯ These authors contributed equally to this work.
* kory.johnson@tuwien.ac.at

**Data Availability Statement:** All code, parameters, data, and corresponding citations are available at https://github.com/korydjohnson/COVID_heterogeneity.

## Abstract

In light of the continuing emergence of new SARS-CoV-2 variants and vaccines, we create a robust simulation framework for exploring possible infection trajectories under various scenarios. The situations of primary interest involve the interaction between three components: vaccination campaigns, non-pharmaceutical interventions (NPIs), and the emergence of new SARS-CoV-2 variants. Additionally, immunity waning and vaccine boosters are modeled to account for their growing importance. New infections are generated according to a hierarchical model in which people have a random, individual infectiousness. The model thus includes super-spreading observed in the COVID-19 pandemic which is important for accurate uncertainty prediction. Our simulation functions as a dynamic compartment model in which an individual's history of infection, vaccination, and possible reinfection all play a role in their resistance to further infections. We present a risk measure for each SARS-CoV-2 variant, $\rho^\nu$, that accounts for the amount of resistance within a population and show how this risk changes as the vaccination rate increases. $\rho^\nu$ highlights that different variants may become dominant in different countries—and in different times—depending on the population compositions in terms of previous infections and vaccinations. We compare the efficacy of control strategies which act to both suppress COVID-19 outbreaks and relax restrictions when possible. We demonstrate that a controller that responds to the effective reproduction number in addition to case numbers is more efficient and effective in controlling new waves than monitoring case numbers alone. This not only reduces the median total infections and peak quarantine cases, but also controls outbreaks much more reliably: such a controller entirely prevents rare but large outbreaks. This is important as the majority of public discussions about efficient control of the epidemic have so far focused primarily on thresholds for case numbers.

## Introduction

The continued waves of the COVID-19 pandemic present unique challenges to regulatory bodies and governments. At issue is the balance between restricting behavior in order to

**Funding:** AG was supported by FWF Austrian Science Fund grant Y00782. JP has received funding from FWF Austrian Science Fund project P32896-B. The funders had no role in study design, data collection and analysis, decision to publish, or preparation of the manuscript.

reduce the spread of SARS-CoV-2 and the desire to return to a normal state-of-affairs. On one hand, many countries provide a deluge of statistics to measure the severity of COVID-19, even on a highly granular level. These statistics then inform complex decisions on how many restrictions to enforce. On the other hand, some countries lack sufficient testing to accurately track the spread of COVID-19. Our guiding question is, what statistics should be considered when determining if mitigation measures should be increased or decreased? Of concern is the oft-repeated scenario in which a new variant emerges which spreads more rapidly either due to increased infectiousness or vaccine escape.

To this end we create a compartment model that has compartments for each vaccinated, infected, and recovered group (for each variant), and add dynamic interactions between these groups as well as immunity waning and boosting. For example, someone could have been infected with the original SARS-CoV-2 variant, then receive a vaccine, then perhaps later become infected with a new SARS-CoV-2 variant. The resistance to further infection conferred by such a history is distinct from those who have, for example, only been vaccinated. These factors influence the effective reproduction number: the expected number of new infections caused by a currently infected individual. We can then simulate infections using this model in order to answer questions about case dynamics when a new SARS-CoV-2 variant is introduced. Each infectious compartment generates new infections according to the momentum model of [1], which accounts for superspreading.

We also add a controller to our simulations, which can both observe and intervene in the compartment model. The controller is thought of as a governing agent which is responsible for both keeping COVID-19 outbreaks at manageable levels and not imposing unnecessary restrictions, i.e., for keeping outbreaks under "control." In order to mimic a real-life entity such as a government, the controller must be constrained in various ways. First, the controller does not observe latent variables such as infectiousness, only raw data such as number of new cases (for each variant), which account for only a proportion of true infections equal to the detection ratio. Second, these statistics are observed with a lag, i.e., there is a delay between when infections occur and when cases are observed. Third, the controller uses non-pharmaceutical interventions (NPIs) such as mask wearing, testing and tracing, and gathering restrictions (soft lockdowns)—which mitigate the effective reproduction number of SARS-CoV-2. Lastly, these interventions cannot change continuously: there is a mandatory temporal gap after an intervention before the controller can intervene again.

Under these constraints, we are able to explore what statistics the controller needs to respond to in order to effectively suppress new outbreaks. Note that we are not advocating a particular intervention or comparing their efficiency [2, 3]. Instead, we are considering what information could best inform timely decisions on modifying NPIs and assess the efficacy of the different controllers. We note that the success of the controller we implement will not be mirrored exactly in reality: few if any governments willingly act as rapidly as stipulated, and people modify their behavior in response to the perceived severity of the outbreak [4]. Notably, immediate action is often not taken when a country crosses the case thresholds stipulated by the World Health Organization [5, 6]. This is exacerbated by the lag between infection and case observation (the "delay" parameter): governments which either fail to collect adequate data or do not make decisions using a forecast will make decisions with greater delay. Yet, we can show that the general principles of efficient control are robust.

First, we provide a summary of the Methods. The Results section then compares the modelled dynamics of the SARS-CoV-2 variants, as well as the controllers' effectiveness in containing both current and hypothesized variants. The Discussion highlights broader implications of our research and further questions which could be explored within the framework. Detailed structured Methods are included after the Discussion, with subsections which describe the

compartment model with different vaccines and variants, the controllers we consider, how waning is implemented, and how the simulation is initialized to mimic a real outbreak.

## Summary of methods

Here we explain the gist of the methods, while all details are given in section Methods at the end of the manuscript.

### Superspreading

We follow [1] to incorporate the variability in individual infectiousness in our model. Such superspreading has been demonstrated to improve the robustness of forecasts by increasing the width of prediction intervals. In contrast to usual compartment models where each individual has the same effective reproduction number $\mathcal{R}_{e,t}$, the momentum model [1] uses individual-specific reproduction numbers which are sampled according to a Gamma distribution with mean $\mathcal{R}_{e,t}$ and variance $\mathcal{R}_{e,t}^2/k$. The superspreading dispersion parameter, $k = 0.1$, corresponds to approximately 10% of infected individuals causing 80% of new infections [7]. The random, individual infectiousness is aggregated each day over all individuals to yield the "momentum" of the disease. The expected number of new infections on a day is given by a weighted sum of these daily momentum values, with weights given by the generation interval, $w_m$. The true infections are then Poisson distributed with this mean (see Eqs (3), (4) and (5) in the Methods section).

### Compartment model and notation

We build a compartment model in where each compartment represents a group of people with a unique history of infection and vaccination given by $h$. Each unique history confers a group-specific resistance to future infection by any variant $\mathcal{V}$. All variants have a basic reproduction number which is proportional to that of the wild type via the coefficient $\lambda^{\mathcal{V}}$ : $\mathcal{R}_0^{\mathcal{V}} = \lambda^{\mathcal{V}} \mathcal{R}_0^{WT}$ With this notation, we can then define the effective reproduction number of variant $\mathcal{V}$ within group $h$, $\mathcal{R}_{e,t}^{h\mathcal{V}}$, as the product of all factors which affect the basic reproduction number of that particular variant at time $t$: $\mathcal{R}_{e,t}^{h\mathcal{V}} = L_t(1 - M_t)(1 - \gamma^{h\mathcal{V}})\lambda^{\mathcal{V}} \mathcal{R}_0$. Here, $L_t$ is the effect of seasonality, $M_t$ is the effectiveness of NPIs on reducing the infectiousness (see Eq (7) of Methods), and $\gamma^{h\mathcal{V}}$ is the resistance of group $h$ to infection with variant $\mathcal{V}$. Resistance parameters are presented visually in Fig 1. Furthermore, we generalize the model to include gradual waning of population immunity which takes the form of individuals randomly transitioning to less-resistant groups over a specified period of time. The Methods section, at the end of the paper, provides details of resistance parameters, dynamic compartment creation, and the implementation of waning.

### Model setup and validation

We initialize the simulation using the history of infections and vaccinations in Austria using data from [8–11] (see Methods: Initializing the Simulation for details). In order to validate our model, we use data available on June 12, 2021, and asses how it fits observed cases into the (then) future. Then, we initialize our model on data available on August 8, 2021, introduce a hypothetical variant we term Omega, and simulate future cases through May 2022. As the variant Omicron appeared in the process of revising this paper, we adjusted the resistance parameters to approximately match what was known about Omicron; yet, we assume the same

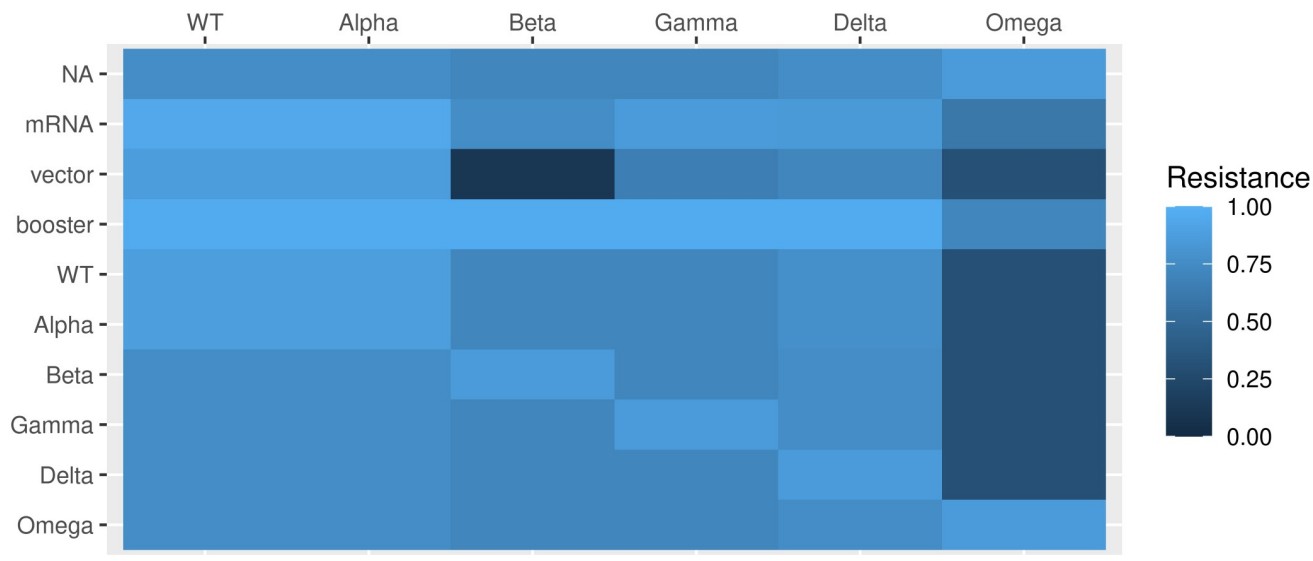

**Fig 1. Midpoint estimates of resistance parameters.** Complete statistics including confidence intervals are given in Table 2.

generation time for all variants (with the exception of Fig F in S1 File), and hence we keep the label Omega for the hypothetical variant to note this distinction.

We specify a detection ratio that an infected individual is diagnosed and appears in the official case statistics; this is important, because the controller responds to detected cases and our validation experiments compare to observed cases. In the application to Austrian data, we use a detection ratio of 1/1.4, which is informed by local models which consider estimates of IFR and use jointly the detected cases, hospitalizations, and deaths due to COVID-19 [12, 13].

## Results

This section presents simulation results for a setting constructed to be similar to that in Austria. This serves to anchor the simulation in a realistic setting, though the high-level results are applicable beyond Austria as well. To aid comparisons to other countries, data are reported as cases/100,000 inhabitants. For a summary of quantitative results, see Table 1.

### Comparing variants

Ultimately, our model requires many parameters to be set which govern the resistance one variant provides to infection from others as well as resistances conferred due to vaccines. There

**Table 1. Summary of quantitative comparisons between reactive and proactive control.** Statistics are provided for the difference in medians (reactive—proactive control) and corresponding 95% confidence intervals as well as the ratio of the 97.5% quantiles of predicted values (reactive/proactive control). The corresponding simulation summary figures are also given for reference. We note that the values are illustrative and are sensitive to both the specifics of the controllers, how the simulations were initialized, and for how long they were run.

| Variant | Waning | Thresholds | Delay | Figure | Total Infections / 100,000 | | Peak Quarantined Cases / 100,000 | |
|---------|--------|------------|-------|--------|------------------|---------|------------------|---------|
| | | | | | Median (CI) | q-Ratio | Median (CI) | q-Ratio |
| Delta | No | High | 7 | Fig 6 | 509.6 (450, 580) | 1.94 | 108.2 (99, 118) | 2.80 |
| Delta | No | Low | 7 | Fig 9 | 53.2 (29, 79) | 1.18 | 16.8 (14, 20) | 2.02 |
| Omega | No | Low | 7 | Fig 10 | 297.1 (236, 361) | 3.35 | 20.9 (13, 29) | 5.49 |
| Omega | No | Low | 21 | Fig 11 | 309.62 (220, 454) | 2.05 | 10.11 (1, 20) | 1.82 |
| Omega | Yes | Low | 7 | Fig 13 | 656.62 (453, 875) | 3.38 | 83.72 (48, 119) | 5.61 |

**Table 2. Model parameters for different variants.**

| Label | WT | Alpha | Beta | Gamma | Delta | Omega | WT,A,B,G,D | Omega |
|---|---|---|---|---|---|---|---|---|
| $\lambda^{\nu}$ | 1 | 1.3 | 1.25 | 1.40 | 2 | 1.55 | 6 months waned | 6 months waned |
| | - | (1.24,1.33) | (1.2,1.3) | (1.22,1.48) | (1.76,2.17) | (1.35,1.75) | | |
| mRNA | 0.94[42, 43] | 0.94[42, 43] | 0.75[44] | 0.85[43] | 0.84[42, 43] | 0.6[45] | 0.5[45, 46] | 0.05[45] |
| | (0.85,0.96) | (0.85,0.96) | (0.7,0.8) | (0.7,0.93) | (0.7,0.86) | (0.5,0.65) | (0.4,0.6) | (0,0.1) |
| vector | 0.86[42] | 0.86[42] | 0.1[47] | 0.65[48] | 0.7[42] | 0.3[45] | 0.4[45, 46] | 0[45] |
| | (0.65,0.93) | (0.65,0.93) | (0,0.55) | (0.6,0.8) | (0.65,0.73) | (0,0.55) | (0.35,0.45) | (0,0.05) |
| booster | 0.96[45] | 0.96[45] | 0.96[45] | 0.96[45] | 0.96[45] | 0.7[45] | 0.85[45] | 0.3[45] |
| | 0.94,0.98 | 0.94,0.98 | 0.94,0.98 | 0.94,0.98 | 0.94,0.98 | 0.6,0.8 | 0.75,0.9 | 0.2,0.4 |
| WT | 0.87[42] | 0.87[42] | 0.7[49] | 0.7[50, 51] | 0.77[42] | 0.3 | 0.4 | 0 |
| | (0.84,0.9) | (0.84,0.9) | (0.55,0.8) | (0.55,0.8) | (0.66,0.85) | (0,0.55) | (0.35,0.45) | (0,0.05) |
| Alpha | 0.87[42] | 0.87[42] | 0.7[49] | 0.7[50, 51] | 0.77[42] | 0.3 | 0.4 | 0 |
| | (0.84,0.9) | (0.84,0.9) | (0.55,0.8) | (0.55,0.8) | (0.66,0.85) | (0,0.55) | (0.35,0.45) | (0,0.05) |
| Beta | 0.75 | 0.75 | 0.85 | 0.7[50, 51] | 0.75 | 0.3 | 0.4 | 0 |
| | (0.65,0.85) | (0.65,0.85) | (0.8,0.9) | (0.55,0.8) | (0.65,0.85) | (0,0.55) | (0.35,0.45) | (0,0.05) |
| Gamma | 0.75 | 0.75 | 0.7[49] | 0.85 | 0.75 | 0.3 | 0.4 | 0 |
| | (0.65,0.85) | (0.65,0.85) | (0.55,0.8) | (0.8,0.9) | (0.65,0.85) | (0,0.55) | (0.35,0.45) | (0,0.05) |
| Delta | 0.75 | 0.75 | 0.7[49] | 0.7[50, 51] | 0.85 | 0.3 | 0.5 | 0 |
| | (0.65,0.85) | (0.65,0.85) | (0.55,0.8) | (0.55,0.8) | (0.8,0.9) | (0,0.55) | (0.35,0.6) | (0,0.05) |
| Omega | 0.75 | 0.75 | 0.7 | 0.7 | 0.75 | 0.85 | 0 | 0.5 |
| | (0.65,0.85) | (0.65,0.85) | (0.55,0.8) | (0.55,0.8) | (0.65,0.85) | (0.8,0.9) | (0,0.05) | (0.35,0.6) |

The first row gives the assumed increase of $\mathcal{R}_0$ relative to wild-type (WT) based on [54]. The rest give the immunity against new infection following vaccination or previous infection. The upper row per label gives the median estimate; the lower row gives a confidence interval. The references are given in the superscript of the median value. See the text for more details.

are three relevant dimensions in which we allow variants to differ: the basic reproduction number $\mathcal{R}_0^V$, immune escape after vaccination, and immune escape after infection. It is important to view these as three separate components, and we note that increasing severity in multiple categories may over-estimate the true risks of different variants. We assume that the generation interval does not significantly differ among variants, which is consistent with known estimates for Delta and older variants [14–16]. The Supporting Information explores other settings in which variants have generation intervals with lower mean, such as is possible for Omicron.

Table 2 discusses the parameters we use and how they differ between variants. We extend that discussion here by showing how these parameters translate into dimensions of primary concern. To simplify a visual presentation, Fig 2 only shows $\mathcal{R}_0^V$ and vaccine effectiveness, which are the two most important measures that are independent of population composition. We further distinguish vaccine effectiveness between mRNA (○) and vector (Δ) vaccines. Both estimates and their uncertainty are summarized in Fig 2, in which the parameters are drawn from a truncated normal distribution with mean and truncation points given in Table 2. The uncertainty represented in Fig 2 is also included in our simulations.

While Fig 2 summarizes raw parameters, it is not sufficient to characterize which variants are of greater concern within a given population. We also show how these variant characteristics map to variant risk and are affected by the rise of immune resistance due to vaccinations. These are the two dimensions of primary interest: our risk measure combines both the variant profiles and the background population characteristics, while vaccinations provide the long-

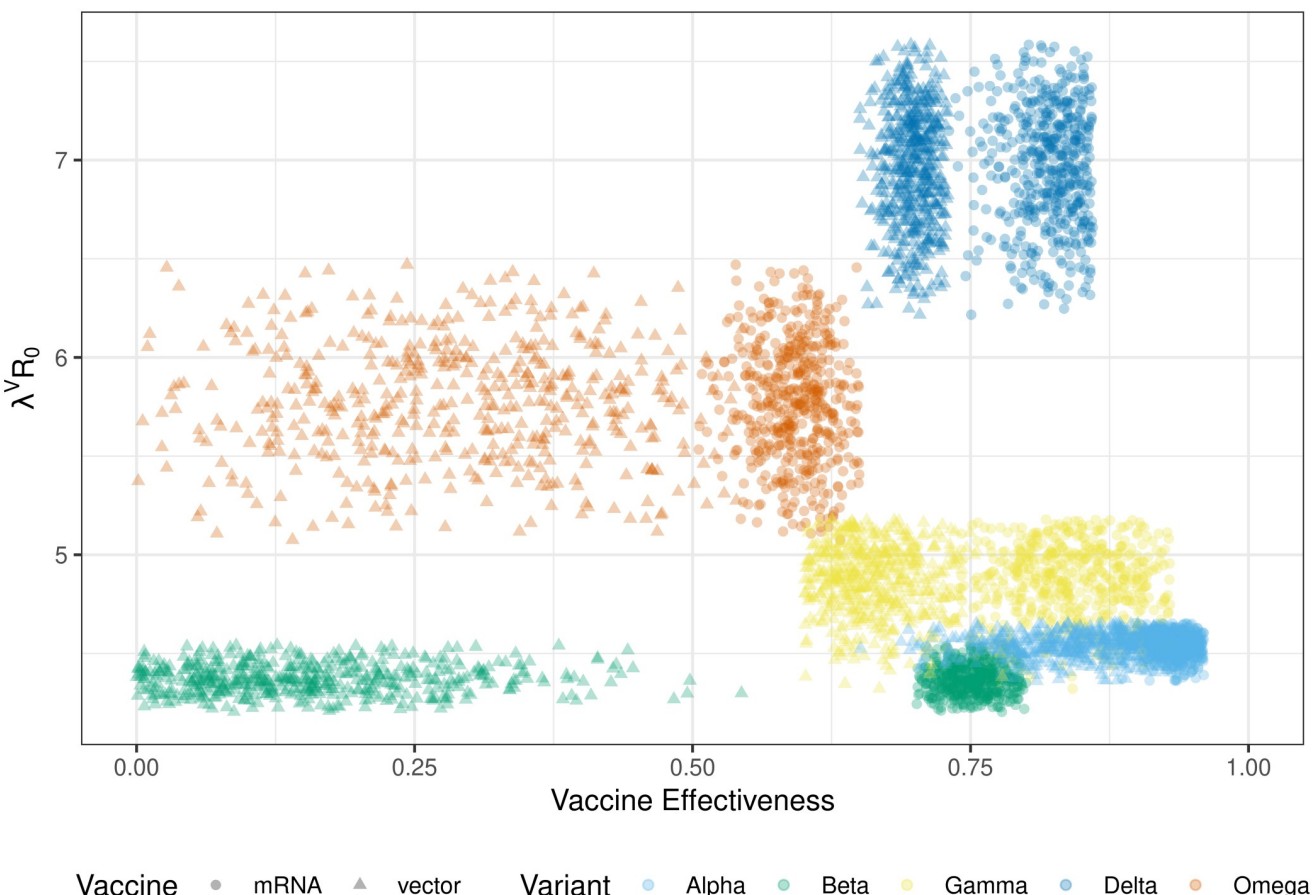

**Fig 2. Basic reproduction number vs vaccine effectiveness for all variants included in our simulations: A(lpha), G(amma), D(elta), and O(mega).** The Omega variant is constructed to analyze hypothetical scenarios. Note that WT is only present in the history of previous infections.

term solution to the pandemic. Therefore, our graphs summarize which variants are of greatest concerns to regions with different vaccination rates. $\mathcal{R}_{e,t}^{h\mathcal{V}}$ from Eq (6) does not easily allow one to compare variants because it specifies both the group that is being infected as well as depends on mitigation and seasonality. Therefore, we remove the time varying components, $(1 - M_t)L_t$, and integrate $\mathcal{R}_{e,t}^{h\mathcal{V}}$ over the susceptible populations $S^h$:

$$\rho_t^{\mathcal{V}} = ((1 - M_t)L_t)^{-1}\mathbb{E}_h[\mathcal{R}_{e,t}^{h\mathcal{V}}] = N^{-1}\lambda^{\mathcal{V}}\mathcal{R}_0^{WT}\sum_{h\in\mathscr{H}}(1 - \gamma)^{h\mathcal{V}}|S_t^h|, \tag{1}$$

where $N$ is the total population size and $\mathscr{H}$ is the set of all infection histories. $\rho_t^{\mathcal{V}}$ is the conceptual driver of outbreaks in our model as it represents the reproduction number of a variant within a particular population by accounting for susceptibility due to immune evasion.

In order to plot $\rho_t^{\mathcal{V}}$ as a function of the vaccination rate $r \in [0, 1]$, we need to consider how the population composition would change and how this would be reflected in the size of our compartments. As we have created a realistic population distribution for Austria on August 8, 2021, we want to maintain this realism over the range of possible infection rates. Therefore, we split the population into two sets: vaccinated and unvaccinated. While descriptively clear, we also give the precise set definitions using the notation introduced in Methods. The unvaccinated cohort $\mathscr{C}_{uv} = \{\mathcal{C}^h \in \mathscr{C} \, s.t. \, \mathbb{N} \cap h = \emptyset\}$ and the vaccinated cohort

$\mathscr{C}_{va} = \{\mathcal{C}^h \in \mathscr{C} \, s.t. \, \mathbb{N} \cap h \neq \emptyset\}$. As vaccines are assumed to be given independently of whether or not someone has been previously infected and recovered, this maintains our population distribution. Let $\mathscr{H}_{uv}$ and $\mathscr{H}_{va}$ contain the partitioned histories corresponding to $\mathscr{C}_{uv}$ and $\mathscr{C}_{va}$, respectively. Lastly, let $N_{va} = \sum_{h \in \mathscr{H}_{va}} |S_t^h|$ and $N_{uv} = \sum_{h \in \mathscr{H}_{uv}} |S_t^h|$ be the size of the vaccinated and unvaccinated susceptible populations, respectively. Note that $N \approx N_{va} + N_{uv}$ as we ignore the comparatively small set of people that are currently infected. We then have

$$\rho_t^{\nu}(r) = \lambda^{\nu} \mathcal{R}_0 \left( \frac{r}{N_{va}} \sum_{h \in \mathscr{H}_{va}} \tilde{\gamma}^{h\mathcal{V}} |S_t^h| + \frac{1-r}{N_{uv}} \sum_{h \in \mathscr{H}_{uv}} \tilde{\gamma}^{h\mathcal{V}} |S_t^h| \right). \tag{2}$$

Observe that Eq (2) is merely a convex combination between $\rho_t^{\nu}$ computed on two different populations: those who are vaccinated (first term) and those who are not (second term). For example, suppose that there is no resistance conferred by previous infection and perfect resistance conferred by vaccination ($\tilde{\gamma}^{h\mathcal{V}} = 1$, $\forall h \in \mathscr{H}_{uv}$ and $\tilde{\gamma}^{h\mathcal{V}} = 0$, $\forall h \in \mathscr{H}_{va}$). In this case, Eq (2) simplifies to $\rho_t^{\nu}(r) = \lambda^{\nu} \mathcal{R}_0 (1-r)$, which is just the basic reproduction number times the proportion of unvaccinated individuals.

Fig 3 shows how $\rho^{\nu}(r)$ changes as a function of the proportion of the population that is fully vaccinated, $r$. In a highly vaccinated population, the Delta and Beta variants are estimated to be similarly infectious, but they diverge significantly in populations with a lower percentage vaccinated. The bands in Fig 3 capture the uncertainty in parameter values shown in Fig 2. The ranking of risks only switches when a large proportion of the population is vaccinated

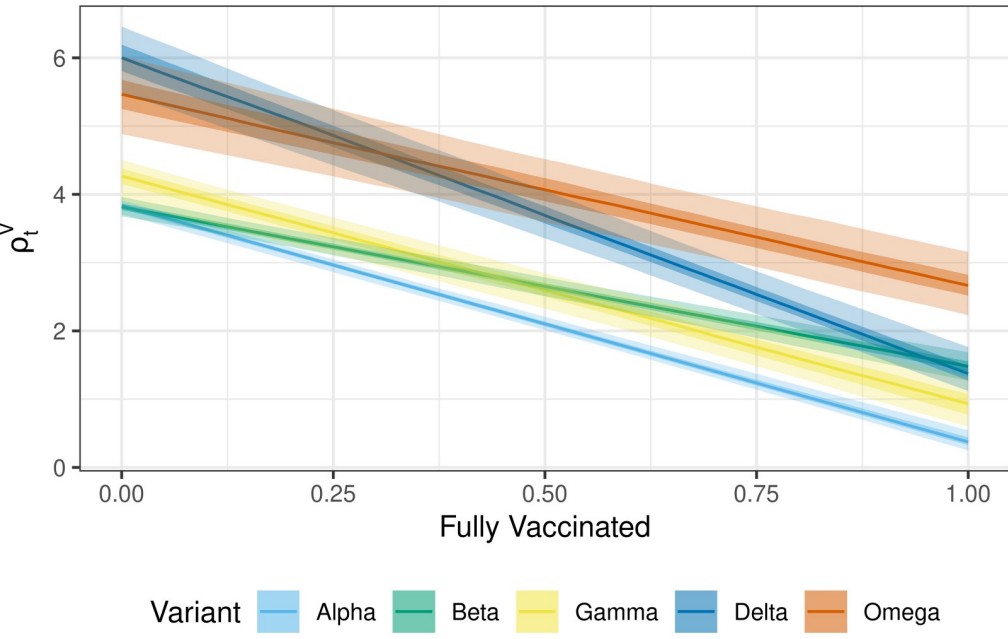

**Fig 3. The reproduction number of variants accounting for immunity within the population, before mitigation and seasonality.** $\rho_t^{\nu}$, defined by Eq (2), is shown as a function of the proportion of the population that is vaccinated. Omega is a hypothetical variant with a higher immune escape: its *relative* advantage thus increases as vaccination level does. Shaded regions correspond to 50% and 95% prediction intervals resulting from the uncertainty in viral parameters summarised in Fig 2. The assumed composition of the population is depicted in Fig 16; it reflects Austria on August 8, 2020, with about 20% of population recovered from infections, mainly by WT (75%), Alpha (22%) and Delta (3%). For simplicity, we assume that vaccination is independent of the infection history. Different population compositions and immunity waning are addressed later (Figs 4 and 12).

(thus increasing the average resistance against variants, which were highly transmissible in a more naïve population). This area is unlikely to be reached without wide-spread and thorough vaccination campaigns. For example, only 88% of the Austrian population is in the 12+ age category, and nearly all of this category would need to be vaccinated to reach the change point.

In order to understand what a future outbreak could look like, we hypothesize a new variant, Omega, with lower basic reproduction number than the currently dominant Delta variant but also higher immune escape. This configuration was chosen in order to create a realistic problem setting that can continually affect regions even after successful vaccination campaigns or after a high number of previous infections.

As Fig 3 uses the vaccine distribution and proportion infected as observed in Austria, it is useful to further demonstrate what $\rho^v(r)$ could look like for other populations with different vaccination campaigns as well as different histories of infections. For example, some countries such as Singapore and New Zealand have had minimal local transmissions and thus little infection-induced immunity. Others, such as the United Kingdom, have experienced higher infection numbers and thus benefit from infection-induced immunity. The behavior of outbreaks in these regions is governed by $\rho^v(r)$ in our models, as this summarizes the combined effect of population make-up and variant characteristics. Thus plotting this statistic for relevant population compositions gives insight into the variants that will be of highest risk in different regions. Fig 4 shows four extremal points for populations and vaccination strategies: 0% vs 40%

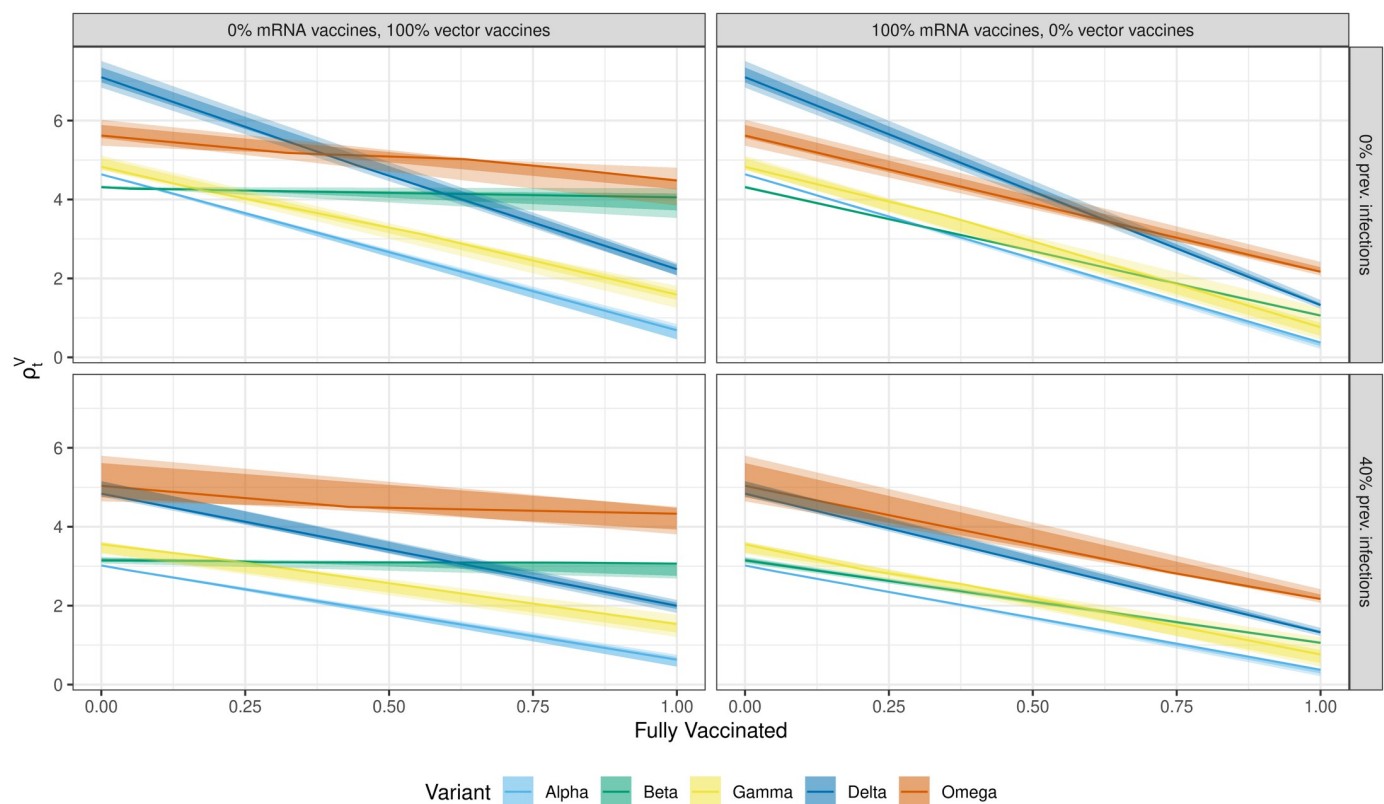

**Fig 4. $\rho_t^v$ accounts for a population-specific reproduction number of each variant.** The charts represent the four vertices of the simplex of vaccine/infection population space. Top row assumes no previous infections; bottom row assumes 40% have acquired immunity due to previous infections. The variants of previous infections are assumed to be distributed according to the estimated proportions in Austria on August 8, 2021, given in Fig 16. We assume that vector vaccines (left) confer lower resistance against infection than mRNA vaccines (right); see Table 2. Shaded regions correspond to 50% and 95% prediction intervals resulting from the uncertainty in viral parameters summarised in Fig 2.

previous infections and 100% mRNA vaccine usage vs. 100% vector vaccine usage. As Eq (2) is linear, any point in-between these extremes can be faithfully represented as a linear interpolation between the graphs in Fig 4.

For example, a country with an epidemic and vaccination history similar to the UK, is represented in the bottom-left image of Fig 4: it has a high degree of previous infections and has relied heavily on vector vaccines throughout the first half of 2021. We see that Omega is the variant of highest risk essentially throughout the entire domain of vaccination proportion. Furthermore, given that we assume that vector vaccines do not protect well against infections with Omega, this risk barely decreases as vaccinations increase, although we see the ordering of risk change among the other variants as vaccination percentage changes. In the opposite corner, we see a country which has had few previous infections and has mainly used mRNA vaccines, such as Singapore. It is clear from Fig 4 that for low vaccination levels, all variants pose a higher risk than they did to the United Kingdom (as there is no prior immunity due to previous infections), with Delta being considerably riskier than Omega. When vaccination rates are higher than approximately 70%, however, the ordering switches and Omega becomes the riskiest—with the highest $\rho_t^v$.

$\rho^v$ is the conceptual driver of outbreaks in our model as it represents the reproduction number of a variant within a particular population, accounting for its susceptibility due to immune evasion. Furthermore, herd immunity is understood in context of current mitigation. This means that populations with greater acquired immunity (lower $\rho^v$) can use fewer NPIs to receive the benefit of dissipating epidemics. Populations with higher $\rho^v$ will need either need higher NPIs in order to suppress an outbreak, or require very strict border controls to prevent importing and allowing community spread of a high-risk variant.

## Controller types

One long-standing question has been how best to control COVID-19 outbreaks when they arise. This subsection explores which statistics should be considered when determining whether to increase or decrease mitigation measures, particularly after the introduction of a new SARS-CoV-2 variant with higher effective reproduction number. Two controller types are considered that either respond to increases in case numbers ("reactive" control) or to increases in an estimate of the reproduction number ("proactive" control). We find that using an estimate of the effective reproduction number in addition to case numbers is a much more efficient strategy.

Responding to the effective reproduction number further helps to address a potential endogeneity effect due to increasing prevalence of COVID-19, i.e., individuals may modify their behavior when the situation either worsens or improves. Most public reporting discusses solely case numbers, so it may be reasonable to assume that individuals base decisions more on either absolute case levels or strong increases in cases. This can be problematic at the start of a wave if case numbers are incredibly low: purely measuring absolute increases or case-number thresholds may trigger a response too slowly. On the other hand, as the reproduction number is not given the same attention in the media, individuals may not change behavior dramatically when it changes. This helps connect NPIs to simulation dynamics, as individuals are not also responding to the same statistics as our NPIs.

As mentioned above, we consider two specific control settings. The first, termed "reactive", only responds to case numbers. There is an upper bound, above which stricter NPIs are used, and a lower bound, below which NPIs are relaxed. This is crafted to mimic the EU protocols for measuring the riskiness of non-essential travel, which assign color codes to regions depending on their publicly reported epidemiological data such as 14-day rate of cases, deaths,

and/or tests administered [6]. Being below the lower boundary corresponds to being "green" whereas above the upper corresponds to being "red". In our results, we show a reactive controller with two different sets of thresholds. The first set, referred to as "low-positivity", uses a lower bound of 25 cases per 100,000 and an upper bound of 150 cases per 100,000 (both measured over a 14-day period). This corresponds the recommendations for a low test-positivity region. The second set, referred to as "high-positivity", uses also a lower bound of 25 cases per 100,000 but an upper bound of 50 cases per 100,000 (both measured over a 14-day period). This is far stricter and corresponds to recommendations for a high test-positivity region.

The second control, termed "proactive", responds to both case numbers and an estimate of the effective reproduction number $\hat{\mathcal{R}}_{e,t}$ given in Eq (14). The same thresholds for case numbers are used as by the reactive control. There is also an upper threshold of 1.2 for $\hat{\mathcal{R}}_{e,t}$. The decision rule is as follows: increase restrictions if either $\hat{\mathcal{R}}_{e,t}$ or cases are above the corresponding upper threshold. Conversely, restrictions are reduced if both $\hat{\mathcal{R}}_{e,t} < 1$ and cases are below their lower threshold. In all other instances, make no changes. For both controllers, when mitigation is modified, it is assumed that a 20% change in mitigation is made (both when strengthening and relaxing NPIs).

**Simulation and control of Delta.** We simulate two main scenarios corresponding to both the growth of a new dominant variant and projections for infections after the variant permeates the population. For concreteness and validation in this section, the new variant is the Delta variant. We simulate infection trajectories using initial conditions when Delta accounts for 20% of current cases. This simulation provides two benefits: first, it provides valuable model validation by starting the simulation when 20% *was* accurate for Austria and comparing simulations to observed cases and statistics; second, it furnishes a sample case for what can happen when a new variant with similar transmissibility advantage reaches this threshold.

The simulation starts on June 12, 2021, as that corresponds to the AGES estimates for Delta prevalence in Austria (see Fig G in S1 File). The same initialization process was used as discussed previously, merely until June 12 instead of August 8. All other parameters needed to initialize our simulations are also taken from observed data on this day. This includes history of new cases, the proportion of population that is vaccinated or previously infected, etc. Fig 5 shows that our model accurately forecasts the proportion of Delta cases as measured by AGES: this holds true regardless of whether a proactive or reactive control is used, as seen in the bottom panel Fig 6. The result is independent of the controller as the controller affects all variants equally.

Simulation results for the low-positivity thresholds are shown in Fig 6, which shows daily incidences (case numbers), the effective mitigation level $\tilde{M}_t = (1 - M_t)$ (reduction in $\hat{\mathcal{R}}_{e,t}$ due to NPIs), and the current estimate of the effective reproduction number $\hat{\mathcal{R}}_{e,t}$. These additional graphs can be used to more precisely monitor both the simulated COVID-19 epidemic as well as the control process. The first observation from Fig 6 is that cases under the reactive control correctly match observed cases around the first peak in mid September. Its intervention history roughly corresponds to Austria's, which loosened restrictions slightly over the summer. Notably, the reactive controller begins increasing restrictions around this peak whereas Austria did not. We note that the only real data used beyond the June 12 start date is the vaccination schedule.

Both controllers relax restrictions at roughly the same point (early July), which approximately coincides with the start of the Green Pass program for European tourism on July 1. Approximately one month later, however, the proactive control would increase mitigation measures again to prevent the start of a new outbreak. As case numbers were so low during

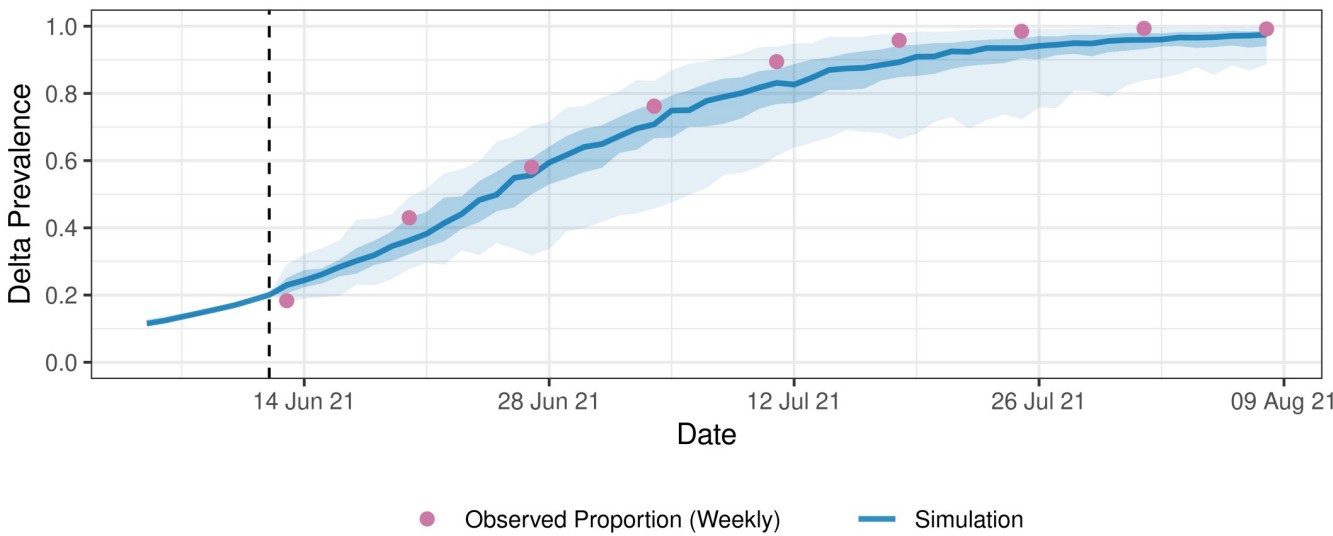

**Fig 5. The changing proportion of new Delta cases is accurately predicted.** The simulation is initialized using information available on June 12, when the observed proportion of Delta in Austria is 20%. Shaded regions correspond to 50% and 95% prediction intervals.

June, merely looking at new cases yields no increase in NPIs for some time. Alternatively, increasing mitigation earlier stabilizes both the effective reproduction number as well as the the number of new cases.

We can also precisely compare both the number of total infections as well as the number of active cases, i.e., those people who would be placed in quarantine. Even if infections are mild and do not require hospitalization, economies suffer if too many people are quarantined (isolated) at any given time. We compute the total number of quarantined cases as merely the sum of new cases over the preceding 10 days. In certain scenarios, more contacts of positive cases may be placed in quarantine, but this in general would just lead to a multiple of the active cases, leaving the conclusions intact. Proactive control not only reduces the median infections and peak quarantine cases, but does so much more reliably: the observations cluster much more strongly around the median. As seen in Fig 7, reactive control has not only higher median values, but also a much more skewed distribution: it is possible to experience massive spikes before the outbreak is brought under control. Given the skewness of the distributions for the reactive controller, difference in medians is measured via a Wilcoxon Rank Sum test.

A more rigorous understanding of the relative efficiency of the controllers is provided by comparing the distributions of the strengths of NPIs used by the two controllers: Fig 8 shows the density of the difference between the observed mitigation and the "balanced" mitigation level, $\tilde{M}_t^*$, defined to be that value for which the effective reproduction number $\hat{\mathcal{R}}_{e,t} = 1$ in Eq (14). Note that $\tilde{M}_t^*$ changes over time due to changes in variant prevalences, seasonality, and acquired immunity in the population. As a reminder, lower values of $\tilde{M}_t$ correspond to stronger NPIs. Fig 8a shows that proactive control trades more time with moderate mitigation for less time at extreme mitigation.

Next, we examine the robustness of these conclusions by using stricter case number thresholds for the controllers as recommended for a higher positivity rate. Fig 9 shows that with stricter case thresholds, simulated cases are much lower because NPIs are triggered more quickly for both controllers. Yet, under a reactive controller, case numbers and the effective reproduction number exhibit a "yoyo" effect, in which they relatively rapidly cycle through periods of

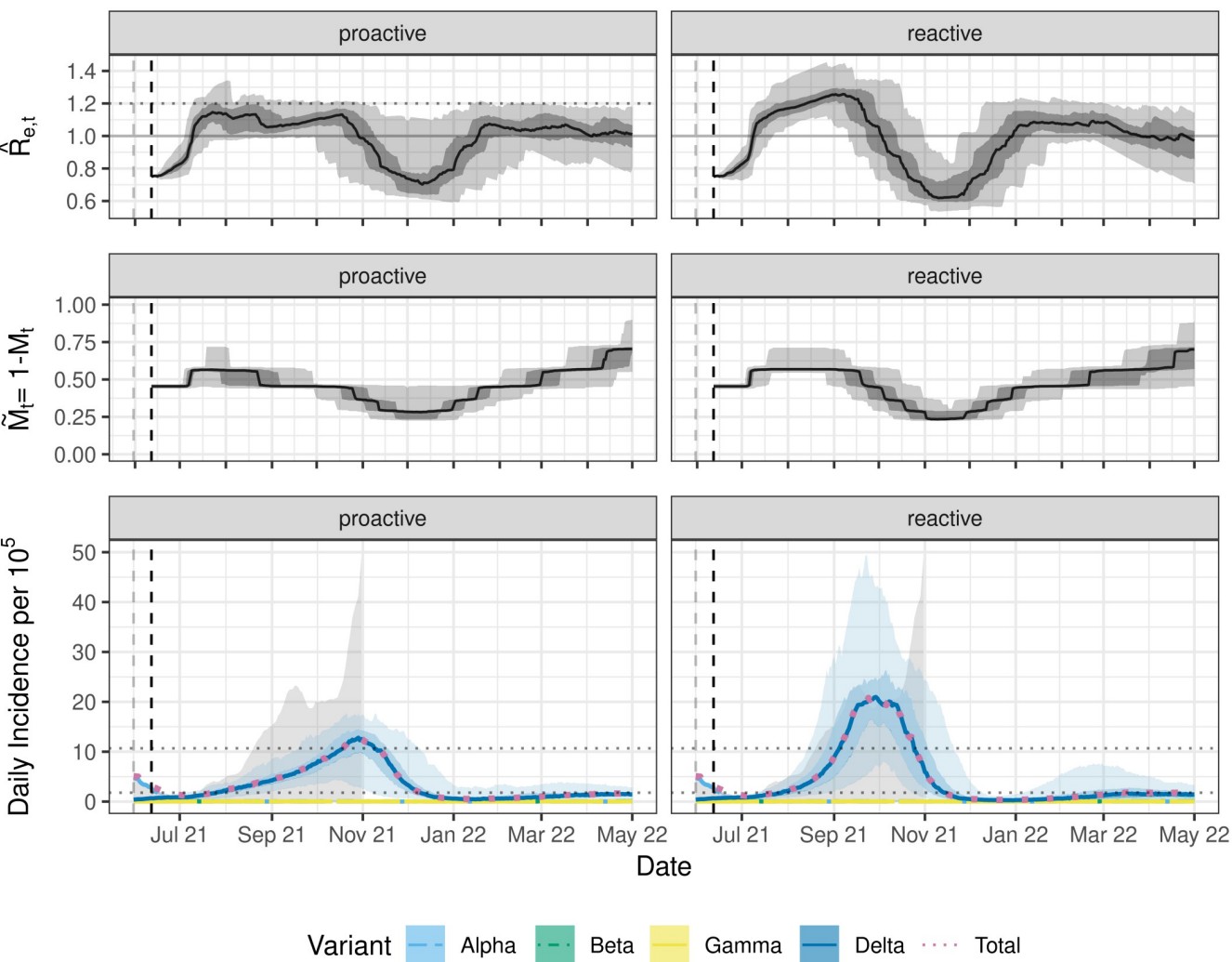

**Fig 6. Responding to the effective reproduction number is more efficient than only using case numbers.** The top row shows the effective reproduction number $\hat{\mathcal{R}}_{e,t}$, middle row the effect of interventions on $\hat{\mathcal{R}}_{e,t}$ (where $\tilde{M}_t = 1$ means no NPIs), and bottom row the daily incidence per 100,000 inhabitants. The simulation starts on June 12, 2021 (black dashed vertical line) when Delta prevalence was at 20% and Alpha was the dominant variant. Initializing the model requires use case numbers from the previous 13 days (gray dashed vertical line). The case thresholds are shown as dotted horizontal lines and coincide with the WHO recommendations for change in NPIs when positivity rate is low [5, 6]); note that the thresholds, 25 resp. 150 per 100,000 within 14 days, are divided by 14 as the y-axis shows daily incidence. The shaded regions gives the 50% (dark) resp. 95% (light) prediction interval. The 7-day moving average of the actual incidence in Austria is given by the height of the gray, shaded area, and data past October is not shown as cases were allowed to dramatically increase under relaxed NPIs. The simulation under reactive control accurately predicts new cases three months after initialization.

increase and decrease. This effect can also be seen for the low-positivity thresholds of Fig 6, but the timescale is much longer. In general, large peaks lead to sufficiently strict NPIs that subsequent peaks appear much later in our simulations. The proactive controller is much more efficient and eliminates this behavior almost entirely. In particular, the left column of Fig 6 shows on average *less strict* NPIs than Fig 9.

The box plots of total infections and peak quarantined cases (see Fig A in S1 File) show that with stricter thresholds, median values are closer together. Reactive control fails to provide reliable control, however, in that some simulated trajectories still produce many infections and quarantined cases. The difference in median total infections per 100,000 is 53.2 (95% CI: 29,

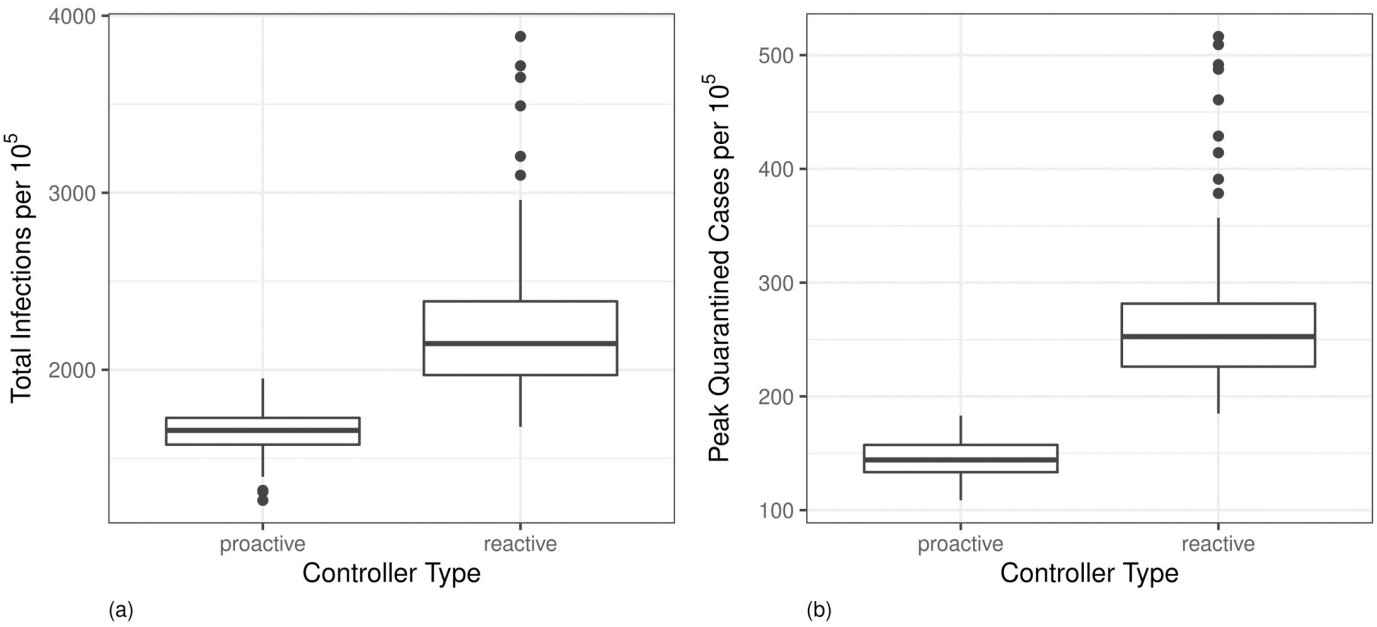

**Fig 7. Proactive control reduces total infections as well as peak numbers in quarantine.** With proactive control, the extent on NPIs reflects not only the active cases but also the estimated reproduction number $\hat{\mathcal{R}}_{e,t}$ (see section Controllers). The difference in medians and corresponding 95% confidence intervals for total infections per 100,000 (over the simulated period) and peak quarantined cases per 100, 000 are, respectively, 509.6 (450, 580) and 108.2 (99, 118).

79) and the difference in median peak quarantined cases is 16.8 (95% CI: 14, 20). That being said, the 97.5% quantile of peak quarantined cases of the reactive controller is twice as high as that of the the proactive one. The mitigation graph for this setting is shown in Fig 8B. Proactive control continues to use a more measured response, whereas reactive control prefers more extreme mitigation levels. In fact, the proactive controller primarily holds $\tilde{M}_t$ slightly below $\tilde{M}_t^*$, which is sufficient to alleviate the epidemic without unnecessarily strict NPIs.

From the point of view of feasibility, the proactive controller makes far fewer interventions than the reactive controller. While a strict reactive controller does keep case numbers low, this results in interventions which occur almost every two weeks. This is the minimum period that we specify in our model as a gap between interventions. It is unlikely that a government would be able to so regularly change policy.

**Simulation and control of a hypothetical variant Omega.** This section introduces a new hypothetical SARS-CoV-2 variant that occupies both a reasonable and empty region of the infectiousness graph in Fig 3. The hypothetical variant, termed Omega, has a lower basic reproduction number than Delta but evades immunity after vaccination or infection by older variants more easily. This provides a scenario in which even a relatively highly vaccinated community will still experience an outbreak and the possibility to explore policies used during the winter of 2021 and spring 2022. Omega is introduced as a weekly import, and for simplicity, one case is imported per day. The distribution of imported infections has no effect on the results, regardless if infections are imported daily or staggered throughout the week. All other parameter and control settings are the same as in the previous subsection on Delta.

While this paper was under revision, variant Omicron has emerged and spread. As it is conceptually similar to our Omega variant, we updated the resistance parameters for Omega to mirror tentative estimates of these parameters for Omicron. Yet the lack of concrete values for other parameters such as mean generation interval and waning of vaccine effectiveness against

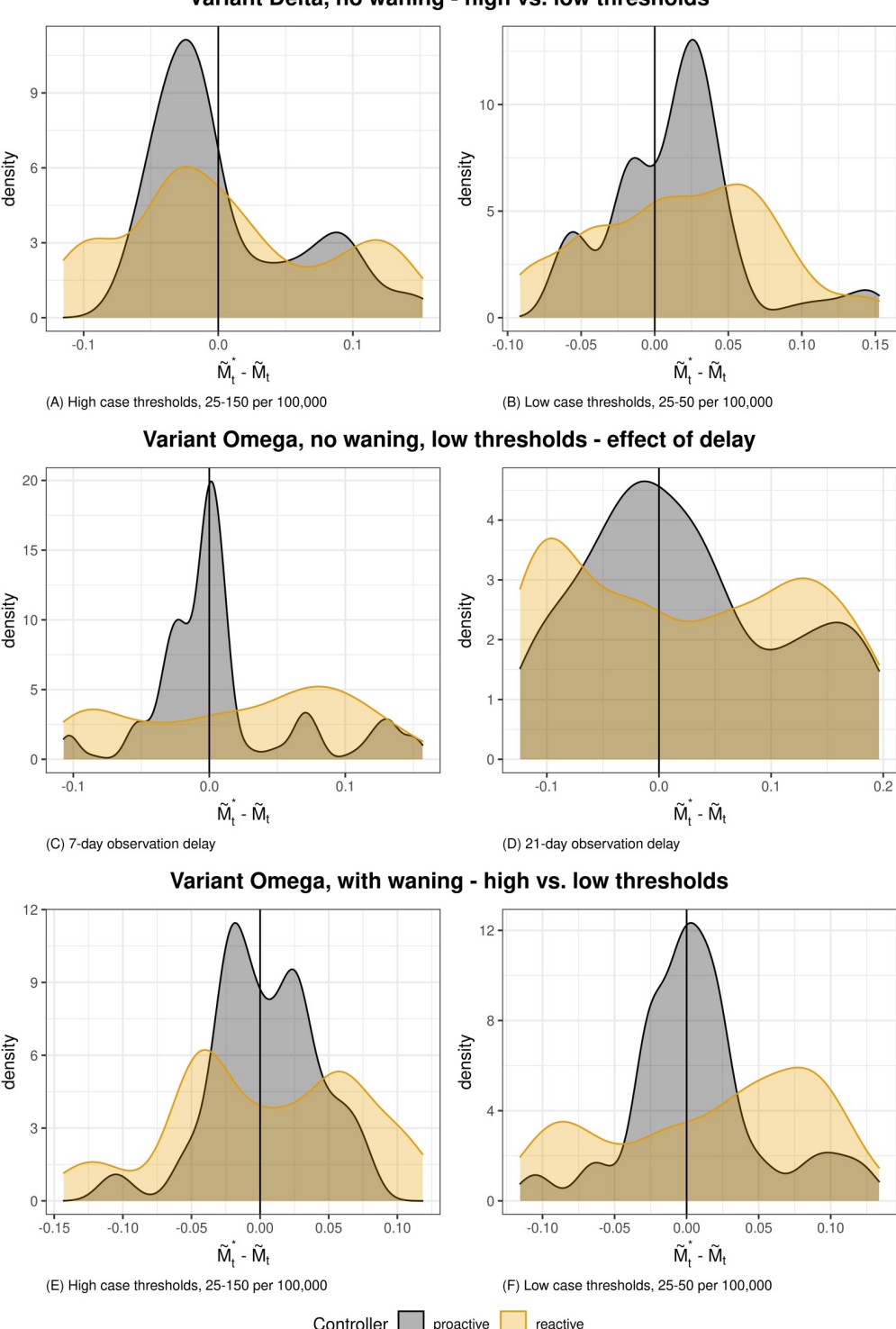

**Fig 8. Proactive control uses mitigation close to the "balanced" level and performs better under information delay.** We define the "balanced" mitigation level, $\tilde{M}^*$, to be that value for which the effective reproduction number $\hat{\mathcal{R}}_{e,t} = 1$. Note that lower values of $\tilde{M}_t = 1 - M_t$ correspond to stronger NPIs. In all settings, proactive control uses mitigation much closer to the balanced value, while the reactive controller compensates for overly lax NPIs by using overly strict NPIs. The degradation in information by using a larger delay (middle row) is seen in the additional dispersion of the proactive mitigation density, though the modal value is still accurate. The time-courses for these simulations are shown in Figs 6 *vs.* 9 (top), Figs 10 *vs.* 11 (middle), and Figs E in S1 File *vs.* Fig 13 (bottom).

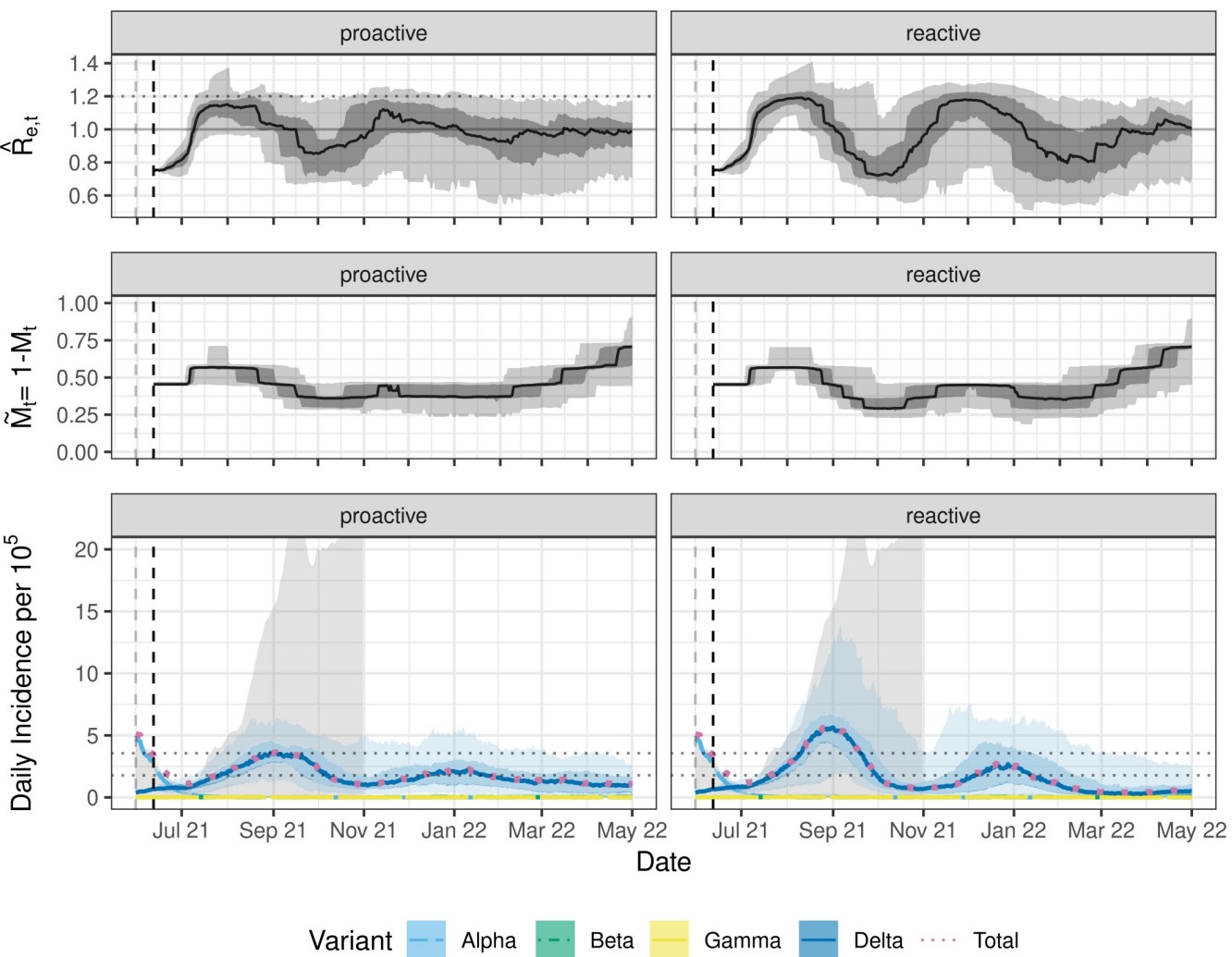

**Fig 9. The proactive control is more efficient than reactive control even when incidence thresholds are stricter.** Stricter thresholds are used such that NPIs are increased when the incidence is 50 per 100,000 inhabitants over a 14 day period. This creates a "yoyo" effect under reactive control that is effectively prevented with proactive control. All other parameters are as in Fig 6.

infection prevent us from making concrete claims about Omicron. As such, we have maintained the name Omega in order to emphasize the inability to properly mimic Omicron at this point in time.

Fig 10 shows how the controllers manage the new Omega variant using the stricter, high-positivity thresholds (25 and 50 cases/100,000 over 14 days). The first few months of each image look qualitatively the same as those in Fig 9: the increased mitigation in the fall delays Omega from being established. The second simulated wave, however, is driven by Omega due to its immune escape. Given the population vaccination levels and compartment structure in Austria, Omega out-competes Delta when the proportion of vaccinated individuals exceeds approximately 30% (Fig 3), which happened in Austria in mid-April, 2021.

The largest difference between scenarios with and without Omega is the width of the infection prediction intervals for the reactive control. Not only does the reactive control allow larger outbreaks, but it is unable to guarantee that all simulation paths are controlled. In

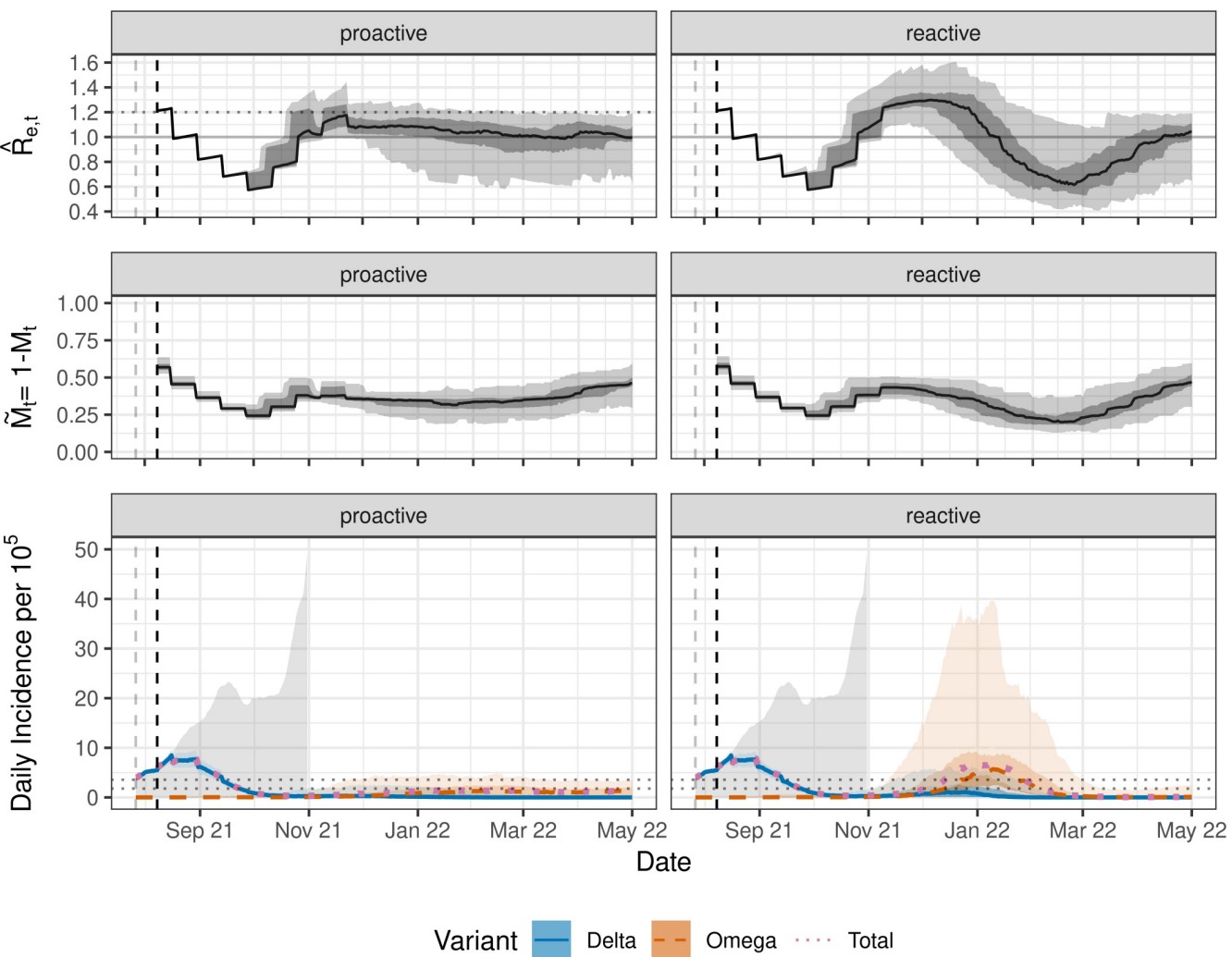

**Fig 10. The reactive control fails to contain new variants that are competitive in highly-vaccinated populations.** The figure shows a scenario with a hypothetical variant Omega that has a smaller basic reproduction number than Delta but has a greater ability to escape immunity post vaccination or infection by other variants (*cf*. Fig 3). Proactive control prevents an outbreak and uses fewer NPIs overall. Other parameters are the same as in Fig 9; note the longer gaps between peaks, which are the result of larger preceding peaks and extended time under more NPIs.

approximately 2.5% of simulations, infections peaked at 40 cases/100,000. The proactive control was able to provide more efficient and reliable control on all simulation instances, effectively preventing an outbreak. This can also be seen by its ability to keep the effective reproduction number stable and near 1. The difference in median total infections per 100,000 is 297.1 (95% CI: 236, 361) and the difference in median peak quarantined cases is 20.9 (95% CI: 13, 29). The difference in the 97.5% quantiles of predicted infections is over 2,000 cases/ 100,000, while for predicted peak cases in quarantine the difference is over 400 cases/ 100,000 (see Fig B in S1 File).

As vaccinations increase, some governments may react to hospitalised (or ICU-hospitalised) incidence instead of case numbers. The rationale is that the controller decides based on hospital capacity, rather than managing the cases. This is particularly enticing as the vaccines reduce severe illnesses or hospitalizations even more than mere infections. Yet this results in a larger delay between infections and actionable information, as there is a larger delay between

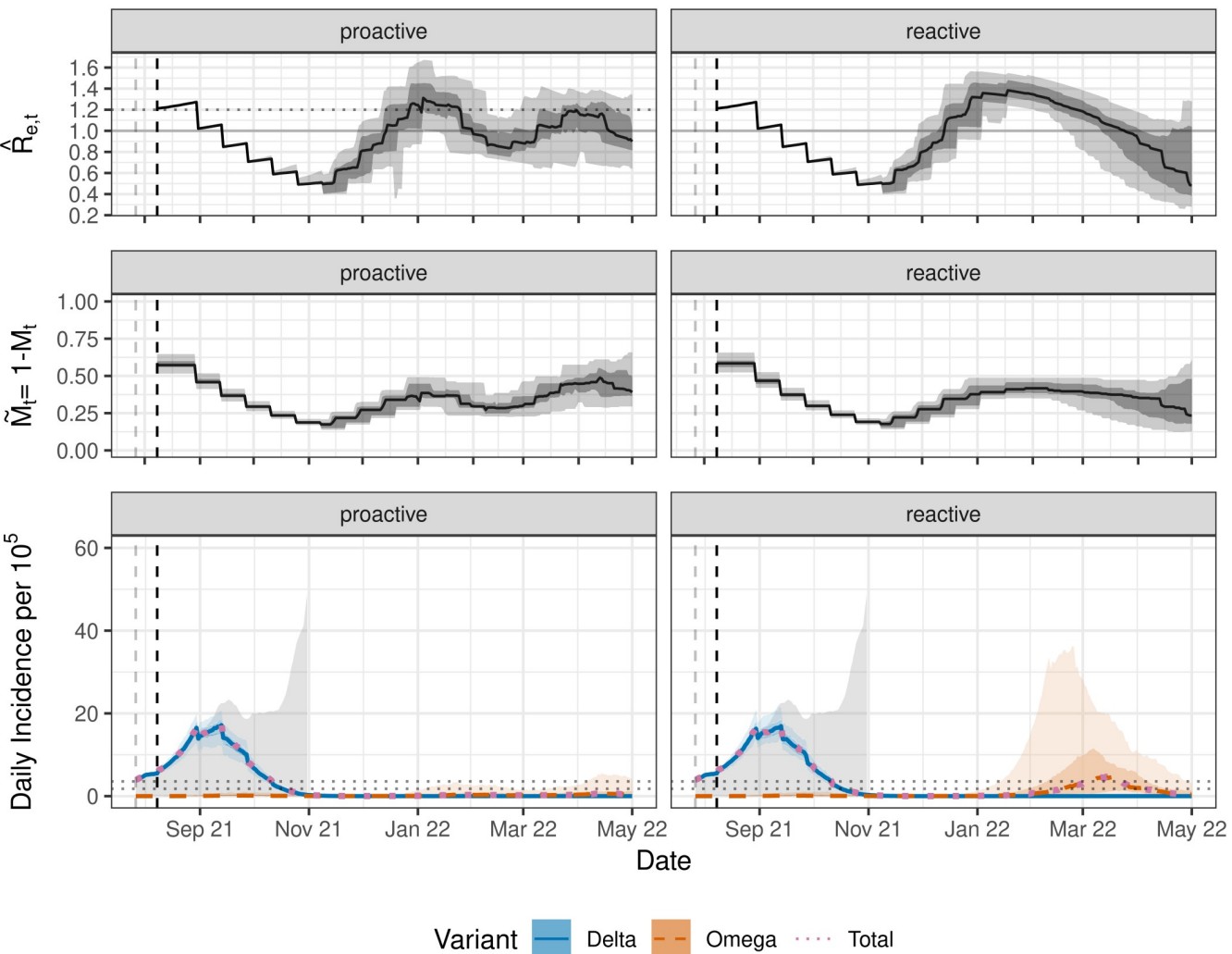

**Fig 11. Controller with a larger delay leads to significantly more pronounced outbreaks, especially with reactive control.** The figure shows a controller with a 3-week delay. A similar delay would be expected when decisions are based on hospitalizations due to COVID-19 rather than on cases. We (conservatively) assume the case thresholds would stay the same. Compare with Fig 10, where the controllers respond with a 1-week delay.

infection and hospitalization. While we do not model hospitalizations as this would require adding an age structure to all of our compartments, we can increase the delay that the controller uses for observing cases. This isolates the effect of the delayed information. If hospitalizations are a constant multiple of infections, then our results translate directly to that domain as well.

Fig 11 shows simulation results with the hypothetical Omega variant, again using the strict, high-positivity thresholds, but with a delay of 21 days (instead of 7 days). In this case, the controller is using the same decision rules to increase or decrease NPIs, but the case data informing this decision is 21 days old. This corresponds to the approximate 2–3 week delay between infection and hospitalization at ICU [17]. We see that the initial outbreaks are significantly more pronounced (*cf*. Fig 10): both controllers begin the simulation by merely increasing mitigation as rapidly as possible. The subsequent outbreak, however, is entirely prevented by the proactive controller, even with delayed information. In fact, the proactive controller using a 21

day delay prevents outbreaks significantly better than a reactive controller with only 7 day delay (*cf*. Figs B and C in S1 File).

One additional effect of delayed information is that restrictions are not lifted in a timely manner. Both controllers maintain the restrictions longer than necessary. This gives rise to the more diffuse mitigation density used by the proactive control as seen in Fig 8C and 8D. For the reactive controller, this has the effect of delaying the Omega outbreak into the spring. In reality, we expect a government to relax sooner, but at the cost of future waves also arriving sooner.

Comparing the difference in peak quarantined cases carries additional meaning in this setting as it is highly correlated with peak hospitalizations. While the medians are again similar, the maximum peak quarantined cases over simulations are nearly twice as high for the reactive controller as for the proactive controller (Fig C in S1 File). Proactive control suffers much less from looking at more delayed data than reactive control. As such, this indicates that looking at rates of change in hospital can be a good indicator, while the delay from reacting to actual hospital utilization is costly.

## Waning and boosting

In the simulation results presented above, populations always increase their immunity to new infections over time, either through vaccination or through infections. In reality, the induced immunity also wanes over time. We focus on simulations with the Omega variant, as it is assumed to be most susceptible to waning. With waning immunity and boosting, the variant risk as measured by $\rho^v$ changes non-monotonically over time (see Fig 12). While boosting rapidly increased in October and November, its strongest effect on $\rho^v$ appears delayed. This is because all vaccinated individuals are eligible for the booster, not just those with waned immunity. The sheer size of the vaccinated groups means many doses need to be administered before the waned groups dramatically decrease in size. The effects of this changing risk profile can also be seen in the infection outbreaks that occur in the simulations and the subsequent controller behavior.

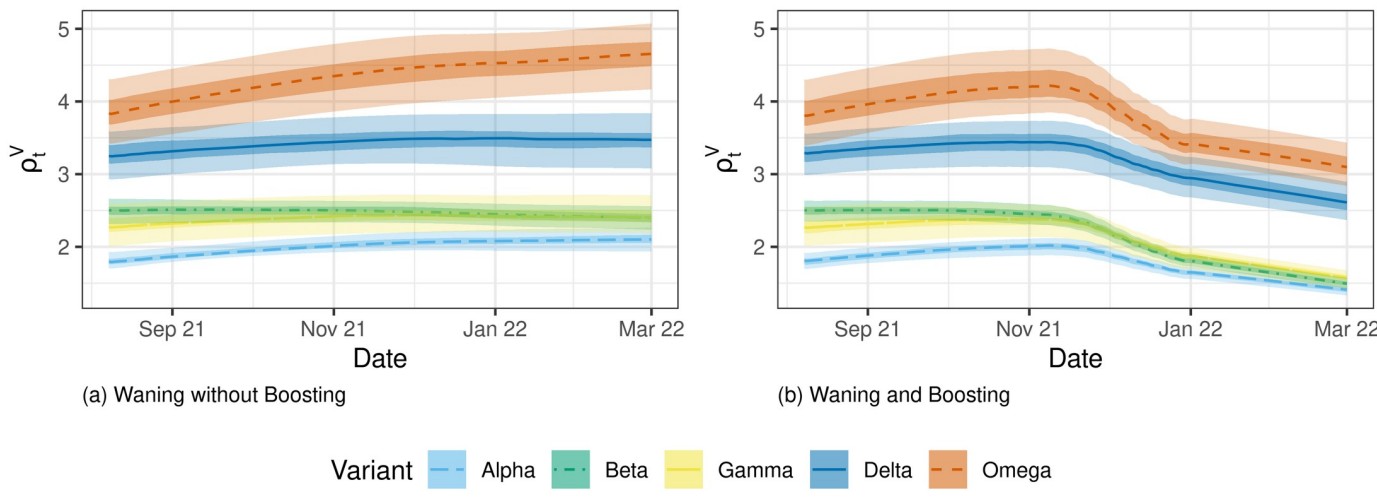

**Fig 12. Waning and boosting change the population-specific reproduction number $\rho_t^v$.** Without waning, $\rho_t^v$ steadily decreases throughout the simulation due to infections and vaccination. (a) Waning of immunity to Omega is strong enough to reverse this effect. (b) Booster shots compensate for immunity waning, and one can clearly see the effect of the wide-scale boosting that rapidly increased in October and November.

Fig 13 shows the simulations with waning but without boosting. Fig D in S1 File demonstrates that adding boosting is sufficient to suppress the Omega outbreak in the winter. For simplicity, we focus only on the strict case thresholds of 25 and 50/100,000/14 days. In the summer, the relative advantage of Omega is too low to see an outbreak in the presence of relatively strong mitigation. Waning immunity then leads to a larger outbreak of the Omega variant than observed without waning in Fig 10, though only under reactive control. The winter outbreak is even suppressed when the proactive controller uses looser, 25–150/100,000/14 days case thresholds (Fig E in S1 File).

In terms of comparison of the controllers, the results of the previous section (without waning and boosting) still hold. There are significant differences in the medians for both total infections and peak quarantined cases between reactive and proactive control. The differences in the upper quantiles of these predicted distributions is even more extreme. We can see in Fig 8E and 8F, that for both high and low thresholds, the reactive controller tends to use extremely high or low mitigation levels, consistently failing to find a balance between suppressing outbreaks and NPI use. In fact, switching to stricter case thresholds exacerbates this problem.

Lastly, we briefly explored simulation dynamics when Omega is modified to spread even more rapidly. In addition to the high immune escape of our baseline Omega variant, we increased its base reproduction number to 2–matching Delta–and reduced the mean of the generation interval from 4.6 days to 3 days (CI: 1.5, 4.5). Such a variant has a significant potential to cause extremely large outbreaks if not controlled efficiently. We confirmed that while the proactive controller suppresses the potential outbreak, the reactive controller does not, even with strict case control thresholds (see Fig F in S1 File).

## Discussion

The model and simulation that we develop provides significant insight into efficient control strategies of COVID-19 outbreaks. The key behavior which we wanted to capture in our simulation was a complex interaction between diverse groups in a compartment model. Our simulation creates compartments for various vaccines, multiple SARS-CoV-2 variants, and specifies well-supported parameters for them all. The interaction rules between compartments are transparent. This allows us to simulate complex scenarios in a realistic and dynamic setting: COVID-19 epidemic spread in Austria. Having an appropriate model is an integral component to controlling outbreaks and a prerequisite for regulating them efficiently [18]. This reduces both human-health and economic costs [19, 20].

It is well known that timely restrictions are more efficient in controlling an epidemic [21, 22]; yet timely decisions can only be taken with a good controller. One can consider addressing this in two distinct ways: 1, use stricter thresholds of a given statistic to motivate change; or 2, use a better statistic, preferably informed by a model of the epidemic [18]. Our study demonstrates that the second type of solution, using the effective reproduction number $\mathcal{R}_{e,t}$ to guide intervention decisions in addition to case numbers, is more efficient and successful at curbing the epidemic. We provide a quantitative comparison of a continuous controller of two types— one which only reacts to cases (reactive), and one which also reacts to the effective reproduction number (proactive). We show that the proactive controller is more efficient and effective at controlling infection outbreaks than the reactive controller, even when it uses data that has a larger delay (such as using $\mathcal{R}_{e,t}$ estimated from hospitalizations). In contrast, the NPIs imposed by the reactive controller are further away from the more efficient "balanced" minimum intervention policy which keeps $\mathcal{R}_{e,t}$ close to 1. By oscillating between over- and under-regulating, the reactive controller fails to provide reliable control of new outbreaks: some simulation paths

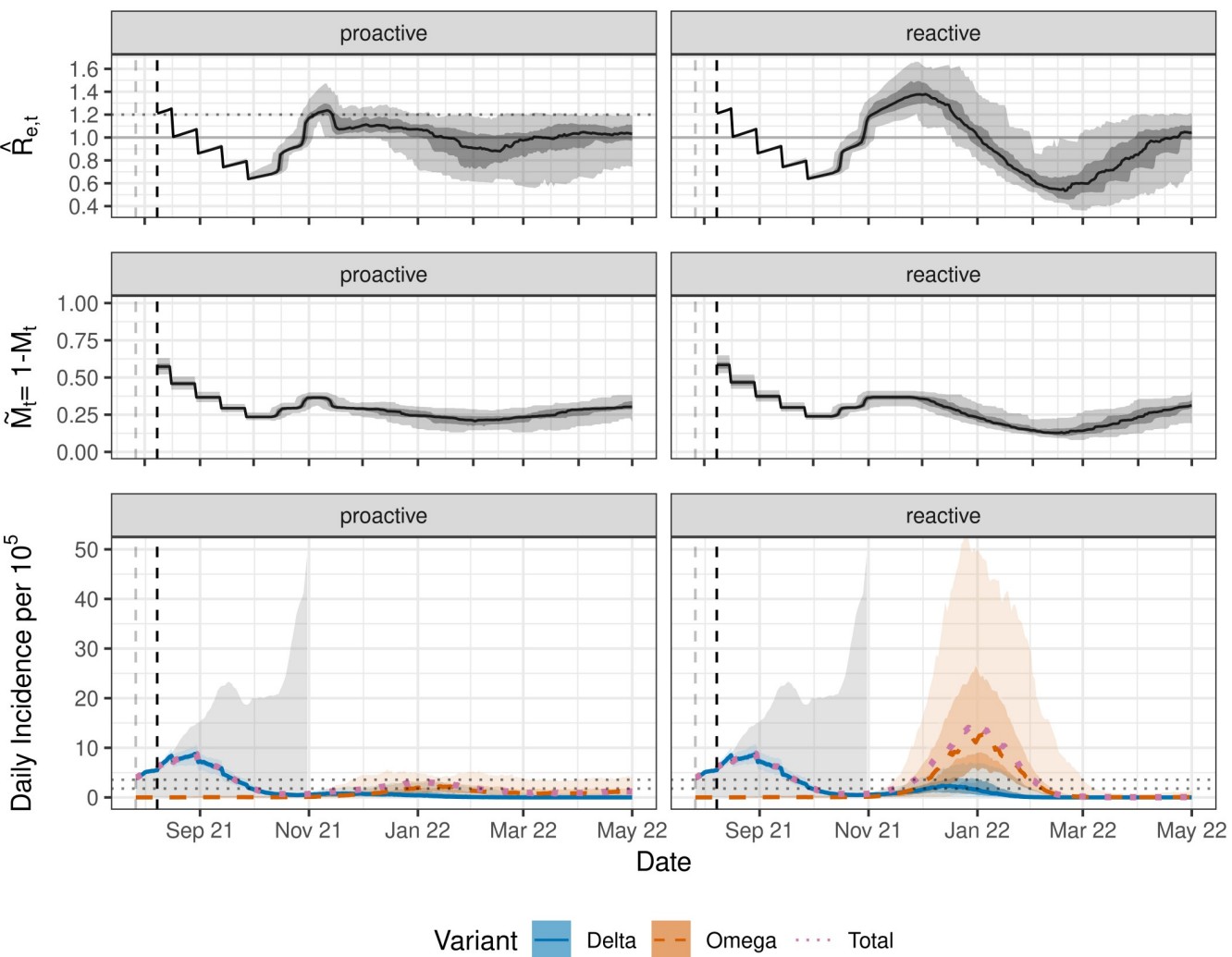

**Fig 13. Waning leads to a larger outbreak of the Omega variant under reactive but not proactive control.** Parameters as in Fig 10, with waning immunity but no boosting.

exhibit large spikes in infections and peak quarantined cases, even when the median values are controlled.

While we specified the NPIs as imposed by proactive *vs*. reactive (governmental) controllers, our framework can be readily generalized to include adaptive human behavior in addition to the NPIs imposed by the controller [4, 23, 24]. A potential limitation of our simulation is our lack of observation level model which would allow us to simulate a fluctuating detection ratio. Although some view positivity-rate to be informative about the detection ratio, we—in the absence of a well-informed model for testing strategies and its saturation—instead consider a range of case-based thresholds which span the gamut of positivity rate scenarios. In all of these settings, we find consistent support for using a proactive control strategy that responds to changes in the effective reproduction number. If the ascertainment bias for detecting cases is changing, one can use $\mathcal{R}_{e,t}$ estimated from the hospitalization rate after taking age structure into account. While inducing NPIs when hospital capacity is reached is a severely delayed

reactive controller and performs poorly, reacting to $\mathcal{R}_{e,t}$ estimated using such delayed information still provides reasonably efficient control as shown in section Controller Types.

Our model does not explicitly incorporate any network structure for individuals' interactions; each individual in the simulation interacts independently with all other members. In reality, infections take place in the household, work, and social environments. The different cross-contamination rates in these environments lead to clusters of observed infections, not only in terms of infections occurring, but also in terms of identifying them via contact tracing. This is partially alleviated by our model accounting for super-spreading. Large super-spreading events are often caused by high infectiousness coupled with a particular network structure. By incorporating super-spreading natively in our model, we are able to produce cluster-like effects due to people that are significantly more contagious than others. Furthermore, other changes in network structure are captured by seasonality or mitigation; seasonality can be caused by increased indoor interaction during winter months, and social distancing rules are common NPIs.

Many parameters need to be specified in our model. These include the basic reproduction numbers of variants, effectiveness of the vaccines against infection, resistance to reinfection (including cross-infection by other variants), and the rate at which resistances wane. A natural question is which of these has a stronger effect on the simulation. The difficulty in answering lies in the distribution of the population across the compartments we describe. Populations with lower vaccination rates but higher rates of previous infection will naturally be more sensitive to cross-infection rates and vice versa. That is why we defined the $\rho^{v}$ parameter, which characterizes a decisive component of the effective reproduction number. We believe $\rho^{v}$ is an important summary parameter: a generalization of the so called vaccinated reproductive rate [25, 26]. It depends on the basic reproduction number as well as the reduction in transmissibility arising from the (partial) immunity acquired by previous infections and vaccinations within a particular population. It determines the relative dynamics of variants as the resistance against them changes within the population. As such, the change of $\rho^{v}$ through time is a key metric to assess the possibility of long-term coexistence of multiple variants [27, 28], in the presence (cross-)immunity which can be waning at different rates.

The paper is not intended to forecast what the future of SARS-CoV-2 will bring: potentially vaccine resistant variants, or variants with even higher base reproduction number, etc. The next important VOC may well have different characteristics to the hypothesised Omega. It is important to keep in mind though, that some VOCs which appear to be out-competed by the currently dominant variant may have a competitive advantage later; this advantage is measured by $\rho^{v}$, assuming that the variants have the same generation interval. The possibility of change in the relative advantage between variants is especially relevant when the emerging variant leads to more severe symptoms. The main claims of this paper, however, hold true for all of these possibilities. Regardless of the process leading to future waves, one certainty is that they will occur. In this eventuality, governments must design methods to identify and react to changes in the pandemic. Our results focus on this common denominator.

## Methods

New infections are assumed to be generated according to the momentum model of Johnson *et al.* [1], which builds upon Cori *et al.* [29]. In the simplest version of the model, new infections $I_t$ are the result of previous infections $I_1, \ldots, I_{t-1}$ via the following recursion:

$$I_t \sim \text{Poisson}(\mathcal{R}_{e,t} \sum_{m=1}^{v} I_{t-m} w_m), \qquad (3)$$

where $\mathcal{R}_{e,t}$ is the time-varying effective reproduction number at time $t$, $\mathbf{w} = (w_1, \ldots, w_v)$ is the generation interval, and $v$ is the maximum number of days for which someone is assumed to be infectious. If $J_m$ denotes the number of people infected by a specific person on the $m$-th day after this person got infected, then we have for $m \in \mathbb{N}$

$$w_m = \frac{\mathbb{E}[J_m]}{\sum_{l=1}^{\infty} \mathbb{E}[J_l]}.$$

We assume that a newly infected individual does not cause secondary infections on the same day, corresponding to $w_0 = 0$, and that infections do not occur after day $v$. The generation interval can be interpreted as the infectiousness profile of infected persons. We set $\mathbf{w}$ to be a discretized gamma distribution with $v = 13$, mean 4.46, and standard deviation 2.63. These are values specific to Austria [30], and are similar to values determined elsewhere [31–33]. We note that the framework is general enough to allow each variant to be specified with a unique generation interval. This is potentially useful for modeling future variants. For example, the expectation of the generation interval of the recently prominent Omicron variant (B.1.1.529) is lower due to shorter latency [34]. This is illustrated in Fig F in S1 File.

The recursion in Eq (3) assumes that all people have the same infectiousness on day $t$. We follow [1] and remove this assumption by explicitly drawing an infectiousness parameter for each infected person from a fixed Gamma distribution with dispersion parameter $k < 1$. This generalization allows for superspreading: the phenomenon of extreme heterogeneity in infectiousness. We set $k = 0.1$, which corresponds to a setting in which 10% of infected individuals cause 80% of new infections [7]. This is an integral component of the difficulty of controlling COVID-19 outbreaks. Individual infectiousness can be aggregated over the infected, resulting in the following process of new infections:

$$I_t \sim \text{Poisson}\left(\sum_{m=1}^{v} \theta_{t-m} w_m\right) \qquad \text{where} \tag{4}$$

$$\theta_s \sim \text{Gamma}(I_s k, \text{ rate} = k/\mathcal{R}_{e,t}). \tag{5}$$

We generalize this model further in order to study the effect of combinations of variants, previous infections, vaccination strategies, and NPIs—and interactions between them—on the effective reproduction number $\mathcal{R}_{e,t}$. This is done by decomposing $\mathcal{R}_{e,t}$ into many constituent parts which depend on the compartments in our model. In order to describe this decomposition, we need notation for compartments.

Our model contains a set of compartments $\mathscr{C}$. Each compartment $\mathcal{C} \in \mathscr{C}$ is a group of people with a unique history of infection and vaccination. The history is encoded as a superscript $h$: $\mathcal{C}^h$. The value of $h$ contains both digits and capital letters, where digits correspond to vaccines and letters correspond to different SARS-CoV-2 variants (when possible, the first letter of the variant name). For example, a group with label $h = A1B$ contains people that were first infected with variant $A$ (Alpha), then vaccinated with vaccine 1, then contracted variant $B$ (Beta). For simplicity, the digit 0 is reserved for the compartment that has neither been vaccinated nor contracted SARS-CoV-2 of any form, i.e., $\mathcal{C}^0$. As it will simplify our notation, we let $\mathscr{V}$ be the set of infectious variants: $\{WT(\text{wild-type}), A(\text{lpha}), B(\text{eta}), D(\text{elta}), G(\text{amma}),\ldots\}$. For clarity, we note here that elements of $\mathscr{V}$, denoted by $\mathcal{V}$, are also valid histories $h$, e.g., $h = \mathcal{V}$ indicates those individuals who have only been infected with variant $\mathcal{V}$.

The only characteristics of the history that affect the model are the total set of experiences (vaccines or infections) as well as the final infection, as this determines the variant one is

infectious with. Furthermore, reinfection with the same variant does not confer additional benefit. Hence we can simplify histories to those that have no repeated capital letters. Lastly, it will often be easier to write equations using the set of histories, $\mathcal{H}$, instead of the corresponding set of compartments, $\mathcal{C}$. As $h \in \mathcal{H}$ is the identifier of a compartment, we will also at times call it a compartment for ease of use.

Each group $\mathcal{C}^h$ contains the total number of people with that history, both infected ($I^h$) and recovered ($S^h$), which are subgroups with the same labels: $\mathcal{C}^h = I^h \cup S^h$. The recovered (or vaccinated) subgroup is written as $S^h$ to emphasize that they are again susceptible to infection, though with a resistance parameter depending on $h$ as described below. All are given subscripts $t$, though for consistency with the generating equations, the subscripted groups have different interpretations. $\mathcal{C}_t^h$ and $S_t^h$ contain *all people* on day $t$ with history $h$ and the subset that are recovered, respectively. $I_t^h$ gives the number of *new* infections with history $h$ on day $t$. For simplicity, individuals recover $v + 1$ days after being infected. We assume that when one is experiencing an infection, they cannot become newly infected (or receive a vaccine). With these simplifications, we have $|\mathcal{C}_t^h| = |S_t^h| + \sum_{m=1}^{v} I_{t-m}^h$. Note that notation for $I_t^h$ has been overloaded to either be the set of people with new infections with history $h$ or the cardinality of this set. This provides consistency with the generating Eq (4).

An important aspect of the simulation is that interaction groups are created dynamically. For example, someone in $S_t^h$ can be infected with a SARS-CoV-2 variant $D$ or become vaccinated with vaccine 4. This person then moves from $S_t^h$ to a new group with identifier $hD$ or $h4$, respectively. The dynamic generation of groups goes hand-in-hand with a dynamic change of population characteristics which may require different mitigation strategies. Crucially, the new group $hD$ or $h4$ can have new resistances to infection.

There is a specific $\mathcal{R}_{e,t}^{h\mathcal{V}}$ for all compartments $h \in \mathcal{H}$ and all infectious variants $\mathcal{V} \in \mathcal{V}$, where group $S_t^h$ is susceptible to infection with $\mathcal{V}$ on day $t$. This can be interpreted as the effective reproduction number of variant $\mathcal{V}$ *solely* within group $\mathcal{C}^h$. As data are often reported in terms of relative transmissibility of SARS-CoV-2 variants, each variant $\mathcal{V}$ has a basic reproduction number relative to that of the original SARS-CoV-2 variant given by $\mathcal{R}_0^{\mathcal{V}} = \lambda^{\mathcal{V}} \mathcal{R}_0 = \lambda^{\mathcal{V}} \mathcal{R}_0^{WT}$. With this group-specific notation, we can define a decomposition of $\mathcal{R}_{e,t}^{h\mathcal{V}}$ as

$$\mathcal{R}_{e,t}^{h\mathcal{V}} = \tilde{M}_t L_t \lambda^{\mathcal{V}} \mathcal{R}_0 \tilde{\gamma}^{h\mathcal{V}} \tag{6}$$

where

- $\tilde{M}_t = (1 - M_t)$, where $M_t \in [0, 1]$ is the effectiveness of NPIs at time $t$ (mitigation of infectiousness). $\tilde{M}_t = 1$ corresponds to no mitigation (full infectiousness), whereas $\tilde{M}_t = 0$ reduces new infections to 0.

- $L_t$ is a seasonality factor at time $t$.

- $\tilde{\gamma}^{h\mathcal{V}}$ is the susceptibility of group $h$ to infection with variant $\mathcal{V}$. We consider $\tilde{\gamma}^{h\mathcal{V}} = (1 - \gamma^{h\mathcal{V}})$ where $\gamma^{h\mathcal{V}} \in [0, 1]$ is the *resistance* of group $S^h$ to being infected by variant $\mathcal{V}$. The value of $\gamma^{h\mathcal{V}}$ depends on the unique history $h$.

Similar to other human coronaviruses and influenza viruses [35, 36], it is widely believed that SARS-CoV-2 follows a seasonal transmission pattern in temperate regions with transmissions peaking during the winter. Possible explanations include different viral longevity due to humidity and air temperature [37, 38], reduced host airway immune response in dry winter months [36, 39], and increased indoor interactions during colder months. We model

seasonality, $L_t$, as in [40] via a cosine transform:

$$L_t = 1 + \frac{\epsilon}{2}\left(\cos\left(2\pi\frac{t - t_{peak}}{365.25}\right) - 1\right), \tag{7}$$

where $\epsilon$ is the amplitude size and $t_{\text{peak}}$ is the date when the transmission rate peaks. Following [41], we assume that the seasonal reduction in transmission is 40% ($\epsilon = 0.4$); the lowest transmission rate is set to July 1, while the highest transmission rate occurs on $t_{\text{peak}}$ = January 1. This is in line with the estimates for the general seasonality of other coronaviruses in temperate climates [35].

The resistance parameters $\gamma^{h\mathcal{V}}$ evolve according to two simple rules. First, if a person with history $h$ is given a vaccine (e.g. 1), resulting in history $g = h1$, they have variant-specific resistances equal to the maximum from those provided by $h$ and 1. For example, history $h$ may provide resistance .9 for infection from variant $D$ and .98 for variant $A$. Vaccine 1, on the other hand, provides resistance .95 for both. The resulting resistance for history $g$ is thus .95 for $D$ and .98 for $A$. Resistance parameters are summarised in Fig 1 and are given in detail in Table 2.

Second, if a person with history $h$ is infected by a variant $\mathcal{V}$, resulting in history $g = h\mathcal{V}$, we consider both the resistances of $h$ and those conferred by $\mathcal{V}$. For this new history $g$, the resistance to an infection with SARS-CoV-2 variant $\mathcal{V}'$ is given by

$$\gamma^{g\mathcal{V}'} = 1 - (1 - \gamma^{h\mathcal{V}'})(1 - \gamma^{\mathcal{V}\mathcal{V}'}), \qquad \text{equivalently} \tag{8}$$

$$\tilde{\gamma}^{g\mathcal{V}'} = \tilde{\gamma}^{h\mathcal{V}'}\tilde{\gamma}^{\mathcal{V}\mathcal{V}'}. \tag{9}$$

Thus, the susceptibility to variant $\mathcal{V}'$ declines as a product of the susceptibility to $\mathcal{V}'$ conferred by the history $h$, $\tilde{\gamma}^{h\mathcal{V}'}$, and the susceptibility to $\mathcal{V}'$ conferred by the infection $\mathcal{V}$, $\tilde{\gamma}^{\mathcal{V}\mathcal{V}'}$. Note that repeated infections with the same variant, i.e. $\mathcal{V} \in h$, then resistances are *not* updated.

We note here that interactions between groups are only considered in terms of resistances, not in terms of infectiousness; a vaccinated individual that nevertheless gets infected with variant $\mathcal{V}$ is considered equally infectious as an unvaccinated individual infected with variant $\mathcal{V}$. This is a simplification—in reality, the viral load in infected, vaccinated individuals appears to decline faster (and is hence lower on average) [52]. In addition, for the same nasopharyngeal viral load ($C_t$s), the probability of detecting an infectious virus using cell culture is also slightly lower [53]. While the model can be extended to include some estimate of lower infectiousness for an infected, vaccinated group, we chose not to do so at present; we do not have a quantitative estimate of how reduced infectiousness translates into reduction in transmission probability in real-life settings, particularly as vaccinated individuals may behave differently.

Specifying the infections created by $I^h$ is notationally far more complex than in Eqs (4) and (5). The issue is that $I^h$ is not solely responsible for creating new infections with this same history at time $t$: $I_t^h$. This problem arises even in the most simplistic multi-variant-vaccine setting. Given compartments $\mathcal{C}^0, \mathcal{C}^1, \mathcal{C}^A$, and $\mathcal{C}^{1A}$, consider the effects of infection and vaccination. New infections $I_t^A$ are produced by both $I^A$ and $I^{1A}$ when they infect members of $S^0$ or $S^A$, while new infections $I^{1A}$ are produced when either $I^A$ or $I^{1A}$ infect members of $S^1$ or $S^{1A}$. Vaccine 1 is administered to a random member of either $S^0$ or $S^A$, and adds members to either $S^1$ or $S^{A1}$, respectively. Given these complexities, we provide simple equations that only show the new infections of a specific history. While it is possible to provide equations for the total new infections for a variant $\mathcal{V}$, this would complicate our expressions and amounts to summing over many different compartments that are distinct in our model. Later, Eq (1) provides a variant-specific equation in order to compare infectiousness of variants in a population.

For consistency of notation, our generating equations describe the number of new infections of a variant $\mathcal{V}$ within a compartment $\mathcal{C}^g$. To do so, consider all compartment histories which are infectious with $\mathcal{V}$: $\mathscr{H}^{\mathcal{V}} = \{h \in \mathscr{H}\ s.t.\ h = \bar{h}\mathcal{V},\ \text{for some history}\ \bar{h}\}$. Each $\mathcal{C}^h$ creates new infections in $S^g$ according to Eqs (4) and (5). We then sum over these groups:

$$I_t^{g\mathcal{V}} \sim \text{Poisson}(\tilde{M}_t L_t \tilde{\gamma}^{g\mathcal{V}}|S_t^g|N^{-1}\sum_{h \in \mathscr{H}^{\mathcal{V}}}\sum_{m=1}^{v}\theta_{t-m}^h w_m) \qquad \text{where} \tag{10}$$

$$\theta_s^h \sim \text{Gamma}(I_s^h k,\ \text{rate} = k(\lambda^{\mathcal{V}}\mathcal{R}_0)^{-1}) \tag{11}$$

and $N$ is the total population size, $N = \sum_{\mathcal{C} \in \mathscr{C}}|\mathcal{C}|$.

Observe that $\theta_s^h$ for $h \in \mathscr{H}^{\mathcal{V}}$, depends on $\mathcal{R}_0^{\mathcal{V}} = \lambda^{\mathcal{V}}\mathcal{R}_0$ instead of $\mathcal{R}_{e,t}^{g\mathcal{V}}$. This is because $\theta_s^h$ gives the "native infectiousness" of—and is solely a property of—$I^h$, separate from the interaction between $I^h$ and $S_t^g$ in the environment experienced at time $t$. Similarly, the second sum of the Poisson argument, $\sum_{m=1}^{v}\theta_{t-m}^h w_m$, does not depend on $g$ because it represents the total infectiousness of $\mathcal{C}_t^h$. New infections, however, depend on other groups $h$ that are infectious with $\mathcal{V}$ and the environment through the remaining parameters in Eq (10). If we ignore susceptibility, mitigation, and seasonality, i.e. $\tilde{\gamma}^{g\mathcal{V}} = \tilde{M}_t = L_t = 1$, and the proportion of infected people in the population is small, then $N^{-1}\sum_{h \in \mathscr{C}}\tilde{\gamma}^{h\mathcal{V}}|S_t^h| \approx 1$. In this case, Eq (10) reproduces (4) in the original setting of the momentum model for a single variant [1].

## Controllers

We assume a controller is interested in constraining the process of new infections, $I_t = \sum_{h \in \mathscr{H}}I_t^h$, and can manipulate $\tilde{M}_t$. Importantly, observed cases are distinct from the underlying process of infections, $I_t$, as not all infections are observed. Our controllers observe only a portion of infections as determined by the detection ratio, and then only some days after infection occurred due to the delay between infection and observation. These parameter values are discussed in Section Initializing the Simulation and Section Results.

Changes in non-pharmaceutical interventions (NPIs) are concretely implemented by setting $\tilde{M}_{t+1} = \delta\tilde{M}_t$ in Eq (10). Thus, increase in NPIs such as mask mandates have the effect of a multiplicative decrease in transmissibility. We have explicitly assumed a certain compound *effect* of NPIs rather than specifying them. Our goal is not to prescribe which combination of NPIs to use, but to demonstrate differences in efficiency of containment strategies that result from using different statistics to guide the decision on the timing of the NPIs.

We consider two types of controllers which react to different statistics computed from case data. Both increase and decrease mitigation, $\tilde{M}$, by some proportion $\delta \in [0, 1]$ whenever they intervene. The first controller changes the effect of NPIs when daily cases pass pre-specified boundaries and is termed a "reactive" controller. This is a controller which increases NPIs when reported daily cases are over some high threshold (e.g. 150 per 100,000 over the last 14 days) and decreases NPIs for case numbers below a low threshold (e.g. 25 per 100,000 over the last 14 days) so long as case numbers are not increasing. A second type of controller, termed "proactive", also utilizes an estimate of the effective reproduction number.

Let $\bar{\mathcal{R}}_{e,t}$ be the effective reproduction number given aggregate statistics which ignore compartment history and the type of infection. This is equivalent to taking expectations of our group- and variant-specific $\mathcal{R}_{e,t}^{h\mathcal{V}}$ over individuals in the population as well as the infectiousness of strains in $\mathcal{V}$. The distribution over compartments weights by the size of the susceptible group: $|S^h|$. Similarly, the distribution over variants weights by the current total infectiousness

of the variant in the population: $\sum_{h\in\mathcal{H}^\mathcal{V}}\sum_{m=1}^{v}\theta_{t-m}^{h}w_m$. We have:

$$
\begin{aligned}
\bar{\mathcal{R}}_{e,t} &= \mathbb{E}_\mathcal{V}\mathbb{E}_h[\mathcal{R}_{e,t}^{h\mathcal{V}}] \\
&= W^{-1}N^{-1}\tilde{M}_t L_t \sum_{\mathcal{V}\in\mathcal{V}}\left(\sum_{g\in\mathcal{H}}\tilde{\gamma}^{g\mathcal{V}}|S_t^g|\sum_{h\in\mathcal{H}^\mathcal{V}}\sum_{m=1}^{v}\theta_{t-m}^{h}w_m\right) \qquad \text{where}
\end{aligned}
\tag{12}
$$

$$
W = \sum_{h\in\mathcal{H}}\sum_{m=1}^{v}I_{t-m}^{h}w_m.
\tag{13}
$$

Here, we have ignored the small number of currently infected individuals by dividing by $N$ instead of $\sum_{h\in\mathcal{H}}|S^h|$.

Eq (12) provides a useful summary as an average effect implied by our model which accounts for variant heterogeneity, population diversity, and superspreading. Unfortunately, a controller cannot compute it as it depends on the unknown parameters $\theta^h$, which describe the "momentum" of the disease [1]. A feasible estimator does not consider $\theta_t^h$ to be known, instead using the expected total current infectiousness of $\mathcal{V}$ given $\lambda^\mathcal{V}$ and $\mathcal{R}_0$:

$$
\hat{\mathcal{R}}_{e,t} = W^{-1}N^{-1}\tilde{M}_t L_t \sum_{\mathcal{V}\in\mathcal{V}}\left(\sum_{g\in\mathcal{H}}\tilde{\gamma}^{g\mathcal{V}}|S_t^g|\sum_{h\in\mathcal{H}^\mathcal{V}}\sum_{m=1}^{v}\lambda^\mathcal{V}\mathcal{R}_0 I_{t-m}^{h}w_m\right)
\tag{14}
$$

We note that the statistic above is not meant to be an ideal estimate of the effective reproduction number $\mathcal{R}_{e,t}$, but instead functions as a computationally efficient way to track the spread of infections in a way that is consistent with our simulation framework. While quantities such as $|S_t^g|$ are not known in practice, they can be estimated per variant and vaccine. In fact, this is done when initializing our model and is described extensively in Section Initializing the Simulation along with estimation of (14).

A "proactive" controller changes mitigation measures based on $\hat{\mathcal{R}}_{e,t}$ and case numbers. Behavior is the same as for the reactive controller, except that there is also an upper bound specified for $\hat{\mathcal{R}}_{e,t}$: when the effective reproduction number is higher than this upper bound, restrictions are increased. Reducing restrictions requires $\hat{\mathcal{R}}_{e,t} < 1$ in addition to low case numbers.

## Vaccines and variants

Our model includes two types of vaccines and six SARS-CoV-2 variants. Vaccine types are summarized in two groups: i) mRNA vaccines which include both Pfizer-BioNTech's Comirnaty (BNT162b2) and Moderna's Spikevax (mRNA-1273); and ii) vector vaccines which include AstraZeneca's Vaxzevria/Covishield (AZD1222) and Janssen's COVID-19 vaccine (JNJ-78436735).

We consider six variants: the original wild-type (WT), Alpha (B.1.1.7), Beta (B.1.351), Gamma (P.1), Delta (B.1.617.2) variants, and a hypothetical variant Omega. In general, their relative advantage and effective reproduction number depend on the composition of the population. The first row of Table 2 shows the assumed relative advantage of variant $\mathcal{V}$ over the wild-type, $\lambda^\mathcal{V} = \mathcal{R}_0^\mathcal{V}/\mathcal{R}_0^{WT}$, in a naïve population. The values are computed from recent estimates of $\mathcal{R}_e^\mathcal{V}/\mathcal{R}_e^\mathcal{V}$ using GISAID sequences across different countries which are then aggregated to produce a summary estimate for each variant [54]. In general, this value is thus confounded with acquired advantage due to immunity escape, although the departure appears

small within the time frame of the study (until June 3, 2021). The values are consistent with estimates of $\lambda^{\mathcal{V}}$ assuming substantial immune escape [49, 55] and are on the lower boundary generally accepted for the transmissibility advantage of Delta [56].

We assume that the basic reproduction number of the original strain, $\mathcal{R}_0^{WT}$, is approximately 3.5. There is a wide range of estimates of $\mathcal{R}_0^{WT}$, ranging from about 2 to 6.5 [57, 58]. As $\mathcal{R}_0^{WT}$ depends on interaction networks in a society and will therefore plausibly differ between countries and regions, we use an estimate of $\mathcal{R}_0^{WT}$ from the early Austrian case data [13, p.13], corrected for the assumed seasonality of 40%.

Table 2 summarizes both the assumed effectiveness of the vaccines against infection with SARS-CoV-2 variants as well as the estimates of resistance conferred by previous infection. We use estimates from the recent analysis by [42] (United Kingdom) concerning infections (with RT-qPCR's $C_t < 30$) by the Alpha and Delta variants. While [42] assesses subjects PCR-tested in weekly intervals, it is limited to people younger than 65 years old. We thus extend the lower bound of effectiveness following [43], which also gives estimates for effectiveness of both mRNA vaccines against (symptomatic) infections with Gamma/Beta variants (in Ontario, Canada). We also use estimates from Brazil [48] for the reduction of transmission of the Gamma variant following full vaccination with vector vaccines, and from Qatar [44] and South Africa [47] for the Beta variant. For computational simplicity, we use the resistances after a full vaccination (typically, two doses), and assign this 2 weeks after the first dose of a vector vaccine or 3 weeks for an mRNA vaccine. This is because the error arising from assigning full immunity 1–2 weeks after second dose would be larger than neglecting slightly lower immunity between doses [42, 59–61]. We assume the percentage of people who do not follow up with the second dose (when required) is sufficiently low that it can be ignored.

The reduction of probability of reinfection is based on the estimates by [42] for the WT, Alpha, and Delta variants. We assume that the probability of reinfection is reduced by 87% (84–90%) upon prior infection with the same variant, and it is reduced less (by 77%, (66–85%)) for the Delta variant when the previous infection was by a different variant (typically WT or Alpha). There is less reliable information on the reduction of re-infection for the Beta and Gamma variants. We use the model-based estimate by [49] of 70% cross-immunity (55–80%) for the Gamma variant. Based on the relative sensitivity of the variants to convalescent sera [50, 51]—and in the absence of a direct estimate of reinfection protection for the Beta variant based on a random cross-infection survey—we employ the 70% cross-immunity for the Beta variant as well. The resistances are set to approximately 75% for cross-immunity between (typically rare) combinations where we do not have direct data, and to approximately 85% for reduction in reinfection by the same strain. See Table 2 for exact values. The last two columns show resistances after immunity has waned, which we discuss in the next subsection.

## Immunity waning and boosters

The waning of immunity from both vaccines and infections plays an increasingly important role in the evolution of the COVID-19 pandemic [62, 63]. Our compartment model can be generalized to this setting. We merely need a rule to determine when and how individuals experience waning as well as an additional "intervention vaccine", i.e. booster, given to previously vaccinated individuals.

One constraint is that our simulation does not track individuals, but groups. Therefore, there is no concept of "time since fully vaccinated" that can be used for a gradual waning of immunity. As such, waning needs to be implemented as a transition of people from a susceptible category $S^h$ to a waned category, which will in general be written as $S^{h-}$. In order to account for the different characteristics of vaccine and variant waning, this is represented in two

different ways in our history notation. Infectious variants are still given capital letters, e.g. $W$ and $A$; after waning, however, these are changed to lower case, $w$ and $a$. This changes a group signature from $h = WA$ to $g = Wa$. Note that the individual was infected by $A$ *before* their resistance from $W$ infection waned. Vaccine waning is represented by a minus sign (overloading the general notation from before), e.g. $h = A1$ becomes $g = A1-$. We assume for simplicity that a waned group cannot wane a second time. Therefore our waned vaccine group $A1-$ cannot wane to $a1-$; waning only occurs for the most recent event experienced. We do not consider this to be a significant oversimplification as the first waning event results in the majority of the resistance reduction experienced and the use of booster vaccines returns people to effectively un-waned groups.

The transition between groups happens at a single point in time as opposed to a gradual decrease in immunity. On average, the waiting time before waning is chosen in order to mimic a gradual, population-level decrease in immunity. All individuals in the susceptible subgroup $S^h$ have an exponential waiting time before being moved to the waned group. The waiting time is determined by the number of days, in expectation, until a lower level of resistance is observed. Each day we thus move $\text{Poisson}(S_t^h/180)$ people from $S^h$ to $S^{h-}$, such that, in expectation, all individuals in $S_t^h$ have waned after a fixed waning period of 180 days. Specifically, we assume that over the waning time of 180 days, the resistance to infection by Delta (and older variants) after vaccination with two doses has declined to 40% (35–45%) for vector vaccines and 50% (40–60%) for mRNA vaccines [45, 46]. For the hypothetical variant Omega, we assume both lower initial resistance after vaccination (vector: 30% (0–55%); mRNA: 60% (50–65%)) as well as stronger waning over 180 days (vector: 0% (0–0.05%); mRNA: 0.05% (0–0.1%)). We assume that the immunity post-infection declines similarly as the one after getting a vector vaccine (see Table 2).

The booster shot is implemented as a new vaccine that is only given to people who have been vaccinated, regardless of their waned status. As the real "waning event" is not uniform over the entire population and it is not known whether or not someone has truly waned when given the booster shot, this is a parsimonious representation of the effects. For simplicity, we group all booster vaccine combinations together, such that there isn't any interaction effect between the booster received and any previous infection or vaccination. As the booster is implemented as a vaccine, resistances are updated according to the same rule: the resistance after receiving the booster is given by the maximum, per variant, of current resistance and the resistance provided by the booster. For Delta and older variants, the boosted resistance starts at 96% (94–98%), and wanes only slightly over 6-months, to 85% (75–90%). For the hypothetical variant Omega, we use the tentative values for vaccine effectiveness based on resistance to symptomatic infection with the Omicron variant, and assume boosted resistance starts at 70% (60–80%) and over 6 months, wanes to 30% (20–40%)—see Table 2.

For a simple population consisting of a single group, we have an average resistance behaving as in Fig 14. In practice, many waned compartments that receive the booster will have identical resistances. This is because their resistances $\gamma^{(h-)V}$ are below those of the booster vaccine against all variants $\mathcal{V}$, and because vaccines only raise resistances to a specified level. We do not collapse the compartments, however, so that the history of a compartment can be tracked in subsequent analyses.

## Initializing the simulation

Given the initial conditions, Eqs (10) and (11) control all new infections and vaccine schedules specify new vaccinations. To initialize the simulation, we need to specify an initial collection of compartments $\mathcal{C}$, a vaccine schedule, and a level of NPIs. These are chosen to most accurately

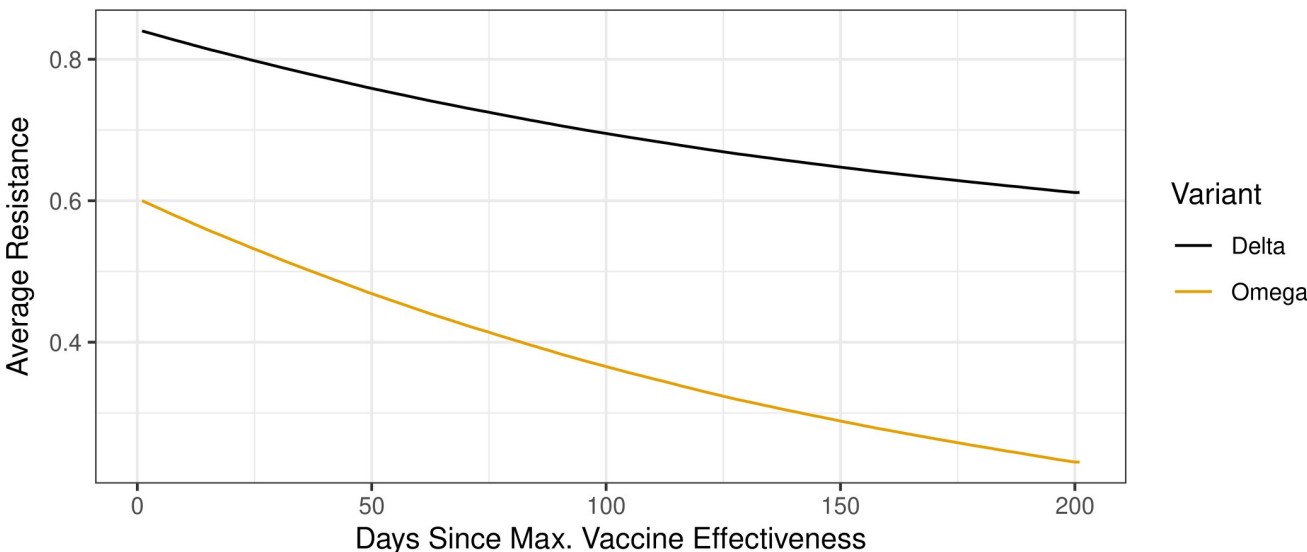

**Fig 14. Average resistance against Delta and the hypothetical variant Omega due to immunity waning in an mRNA-fully-vaccinated population.**

represent the pandemic in Austria. Epidemiological data in Austria is gathered and provided by AGES (Agentur für Gesundheit und Ernährungssicherheit GmbH), the Austrian agency for health and food safety [9]. In addition to tracking the number of cases, hospitalizations, deaths, and tests, AGES tracks the genome sequencing of SARS-CoV-2 samples gathered in Austria to monitor the prevalence of variants of concern (VOC) [10]. The first confirmed case of the Alpha variant in Austria was on January 3, 2021. A VOC sentinel system was subsequently established with one PCR test lab per county submitting a random sample of SARS-CoV-2 specimens for complete genome sequencing.

VOC prevalence in Austria is published online and updated roughly every one to two weeks. Fig 15 provides the historical reported variant prevalences. 2021 has already seen two new variants emerge and quickly become dominant. In Austria, Alpha took approximately 15 weeks between emergence and dominance, whereas Delta took a mere 10 weeks. Both Beta and Gamma variants were observed in Austria, though neither variant made serious headway into the population.

When simulations are initialized for a given date, the relative size of recovered compartments are computed using data from Fig 16. To compute the number of people who were previously infected, we must account for the detection ratio throughout the entirety of the pandemic. Based on [64] and consistent with seropositivity in Austria from mid November [65], we assume that 12% of the Austrian population was infected by the wild-type before January 1, 2021. For 2021, we specify the detection ratio which measures the (age-averaged) probability that an infected individual is diagnosed and appears in the official case statistics. Due to increases in the availability and use of COVID-testing in Austria, we use the following estimates for the detection ratio in 2021: 1/2.3 for January and February, 1/2 for March and 1/1.4 for April and beyond. These are consistent with model-based estimates for Austria which use hospitalizations and deaths to learn about the proportion of unreported infections [12, 13].

We use a vaccine schedule to match that of Austria throughout the simulation period, as the actual vaccination plan is considered to be a background setting of our model as opposed to an intervention by a controller. We use the 7-day median of administered first doses during

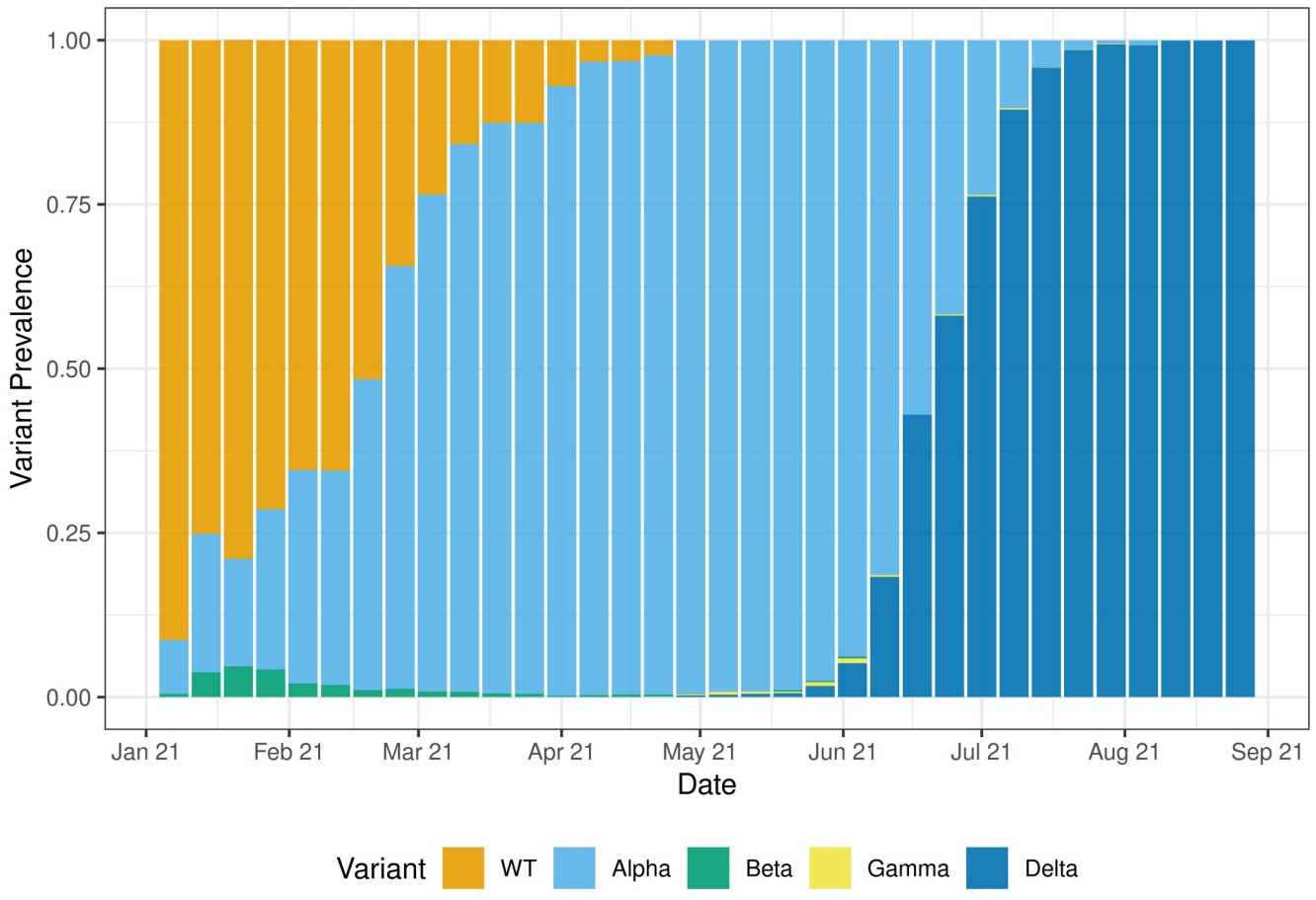

**Fig 15. Reported prevalence of different SARS-CoV-2 variants in Austria in 2021.**

each calendar week [11] up until August 8, 2021. When simulating beyond the window of available data, we use the latest 7-day median and administer this many doses until the expected upper bound on vaccinations is reached. Currently, this bound is 85% of the population. As of August 8, 2021, the distribution of administered vaccines was 72% Pfizer-BioN-Tech's Comirnaty, 10% Moderna's Spikevax, 15% AstraZeneca's Vaxzevria, and 3% Janssen's COVID-19 vaccine. Beyond August 8, the distribution of newly administered doses is 74% from Pfizer, 3% Moderna, 22% Janssen, and 1% from AstraZeneca. The corresponding real data for booster vaccines is used until mid December 2021. Vaccine booster shots began being used in September 2021, with uptake increasing rapidly starting in October and November 2021.

A final step to calibrate our model with current case numbers is to set an initial effect of NPIs. This is done by equating the implied reproduction number from the simulation, $\hat{\mathcal{R}}_{e,t}$, to the observed $\mathcal{R}_e$ in Austria at the time the simulation starts. More concretely, $\hat{\mathcal{R}}_{e,t}$ from Eq (14) is simplified at initialization as our compartment structure features no interaction groups. We assume that all individuals that were previously infected with the wild-type or Alpha are equally likely to have been vaccinated, while people with other infections were not vaccinated as the other variants primarily appeared later. Tacitly, this assumes that those who were previously infected and then vaccinated do not receive an additional benefit due to their initial

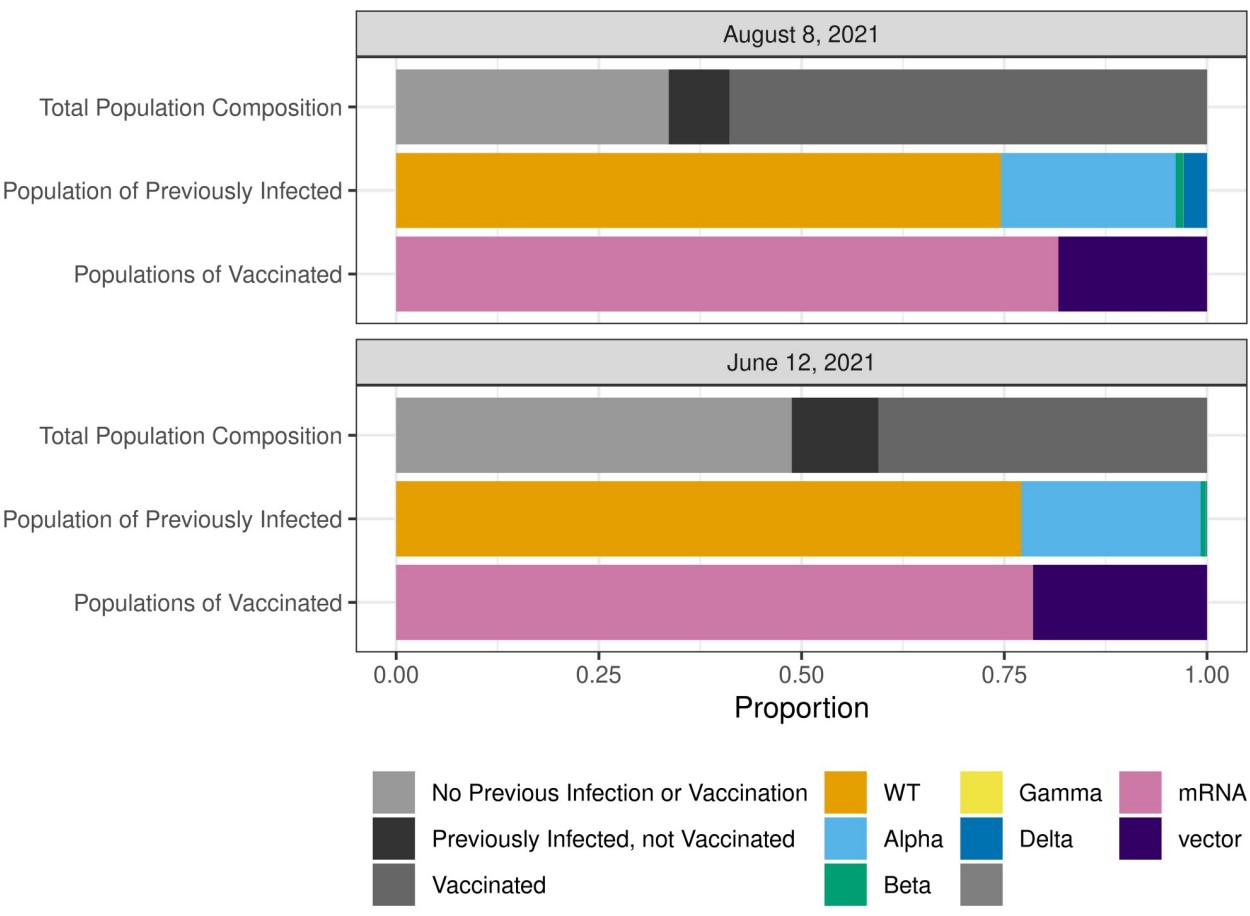

**Fig 16. Estimate of the population composition in Austria on June 12 and August 8, 2021.** Note that while Delta was dominant by August, 2021, the absolute number of infections is comparably low.

infection. This is in line with the update rule for infection followed by vaccination, though could also be considered a result of infection waning due to long-past infections. The number of recovered individuals of each type, $|S_t^h|$, is estimated by taking the cumulative cases for 2021 and scaling by the inverse detection ratio for each time period given above. A proportion of $S_t^h$ for wild-type and Alpha are removed from these groups and placed in the vaccinated groups. This specifies $|S_t^h|$ and $I_{t-m}^h$ for all individual variants and vaccines, allowing us to compute $\hat{\mathcal{R}}_{e,t}$ up to the missing $\tilde{M}_t$ term, which is calibrated to the observed $\mathcal{R}_e$.

In practice, similar steps are required whenever the model is used to simulate outbreaks within a new country or region. As many parameters will not be know with certainty, particularly at the start of an outbreak, the parameters values can be drawn randomly from suitable ranges to capture the underlying uncertainty. This is done for all of our simulations as well. Lastly, the high-level results remain consistent even when the parameter values are not known with certainty: responding to changes in the effective reproduction number is more efficient than merely considering observed case numbers.

## Supporting information

**S1 File.**
(PDF)

## Acknowledgments

We would like to thank Astrid Christine Erber, Joachim Hermisson, Michal Hledík, Pavel Payne and Claudia Zimmermann for insightful discussions and comments on earlier drafts.

## Author Contributions

**Conceptualization:** Kory D. Johnson, Annemarie Grass, Mathias Beiglböck, Jitka Polechová.

**Data curation:** Kory D. Johnson, Annemarie Grass, Jitka Polechová.

**Formal analysis:** Kory D. Johnson, Annemarie Grass, Mathias Beiglböck.

**Funding acquisition:** Jitka Polechová.

**Investigation:** Jitka Polechová.

**Methodology:** Kory D. Johnson, Annemarie Grass, Mathias Beiglböck.

**Project administration:** Kory D. Johnson, Mathias Beiglböck.

**Software:** Kory D. Johnson, Annemarie Grass.

**Supervision:** Kory D. Johnson, Mathias Beiglböck.

**Validation:** Kory D. Johnson, Annemarie Grass.

**Visualization:** Kory D. Johnson, Annemarie Grass.

**Writing – original draft:** Kory D. Johnson, Jitka Polechová.

**Writing – review & editing:** Kory D. Johnson, Annemarie Grass, Daniel Toneian, Mathias Beiglböck, Jitka Polechová.

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
