## [Decision Letter · Decision Letter 0]

10 Nov 2021

PGPH-D-21-00695

Controlling simulated outbreaks of new SARS-CoV-2 variants

Dear Dr. Johnson,

Thank you for submitting your manuscript to PLOS Global Public Health. After careful consideration, we feel that it has merit but does not fully meet PLOS Global Public Health’s publication criteria as it currently stands. Therefore, we invite you to submit a revised version of the manuscript that addresses the points raised during the review process.

We look forward to receiving your revised manuscript.

Kind regards,

Brooke E. Nichols

Academic Editor

Journal Requirements:

1. We ask that a manuscript source file is provided at Revision. Please upload your manuscript file as a .doc, .docx, .rtf or .tex. If you are providing a .tex file, please upload it under the item type ‘LaTeX Source File’ and leave your .pdf version as the item type ‘Manuscript’.

2. Please provide separate figure files in .tif or .eps format only, and remove any figures embedded in your manuscript file.  If you are using LaTeX, you do not need to remove embedded figures.

3. Please note that your Data Availability Statement is currently missing the repository name and/or the DOI/accession number of each dataset OR a direct link to access each database. If your manuscript is accepted for publication, you will be asked to provide these details on a very short timeline. We therefore suggest that you provide this information now, though we will not hold up the peer review process if you are unable.

4. State what role the funders took in the study. If the funders had no role in your study, please state: “The funders had no role in study design, data collection and analysis, decision to publish, or preparation of the manuscript.”

Additional Editor Comments (if provided):

Thank you for your submission on this timely and important issue. As you will see, the reviewers have identified some concerns and opportunities for improvement of the manuscript, which I think you can potentially address with a major revision. I encourage you to revise and resubmit this work to PLOS Global Public Health.

Reviewers' comments:

Reviewer's Responses to Questions

**Comments to the Author**

1. Does this manuscript meet PLOS Global Public Health’s publication criteria? Is the manuscript technically sound, and do the data support the conclusions? The manuscript must describe methodologically and ethically rigorous research with conclusions that are appropriately drawn based on the data presented.

Reviewer #1: Yes

Reviewer #2: Yes

2. Has the statistical analysis been performed appropriately and rigorously?

Reviewer #1: Yes

Reviewer #2: N/A

3. Have the authors made all data underlying the findings in their manuscript fully available (please refer to the Data Availability Statement at the start of the manuscript PDF file)?

Reviewer #1: Yes

Reviewer #2: Yes

4. Is the manuscript presented in an intelligible fashion and written in standard English?

Reviewer #1: Yes

Reviewer #2: Yes

5. Review Comments to the Author

Reviewer #1: Grass and colleagues present a comparison of SARS-CoV-2 epidemic forecasts for Austria under different strategies for imposing MPIs: basing decisions solely on detected cases, versus also taking into account estimated Re.

The technical depth and clarity of the manuscript is outstanding! This is one of the best COVID modeling papers I have read in terms of technical depth and clarity. The inclusion of superspreading dynamics is a strength not shared by most other models. The authors’ choice to deposit source code is a best practice that adds transparency and reproducibility.

Just a few suggestions to help this manuscript have impact for a global/public health audience:

First, the authors should describe how R^e,t (equation 12) is derived in practice. A walk-through of how the authors calculated this using data from Austria would be very welcome. This could help guide the field in how to derive this estimate and incorporate it into decision-making regarding MPIs.

Second, it is not clear whether the authors have incorporated constraints on ascertainment of infections. The “I” term in Equation 12 is not knowable in real-time or with high confidence. Only reported cases are known.

In parts of the manuscript, the words “cases” and “infections” are used interchangeably, but these have different meanings. The controlled should not know the number of infections, only the number of reported cases. Symptomatic cases typically lag infections by approximately the incubation period, and sometimes a longer if there are laboratory delays. Asymptomatic cases depend on frequency of routine testing, e.g., if testing is much less frequent than the duration of detectable positivity, then it will be approximately half the period of detectable positivity.

Third, the “take-home message” in the Abstract and Discussion is that the proactive controller is more effective. This is an obvious result. The proactive controller has access to the same case data and additionally has access to R^e,t data. It seems obvious that providing incrementally more data as input into a controller will always lead to equal or better performance.

Instead, the authors could calculate the % reduction in cumulative infections (and if available, hospitalizations and deaths) and the % reduction in time under different stringencies of MPIs enabled by the proactive controller. Quantitative estimates with 95% CI should be reported in the Abstract and results. This could help the audience understand the quantitative value of adding Re,t data to MPI decision-making.

The finding that the reactive controller recapitulates the epidemic trend in Austria is also worth highlighting as a take-home point. It shows that the decision-making process made by health authorities is well-captured by the reactive controller, which is interesting given that the MPI decision-making process is not always transparent.

Minor comments:

The introduction states that COVID data are available “sometimes even on a county level” – but countries such as the US make this available at an even more granular level such as postal code or census block.

Figure 4 caption typo: extend  extent

Figure 5,6,7,8, 9: need to define the bounds shaded – are these IQR and 95% intervals? Also, need to specify the year and not just the month/day.

Reviewer #2: In this work, Grass et al. developed a transmission simulation model to study the effectiveness of non-pharmaceutical interventions (NPI) in controlling COVID-19 epidemics given different types of responsiveness: Intervention measures are implemented by a controller either proactively (i.e. responds to changes in effective reproduction number (Ne) as well as case positive rates) or reactively (i.e. responds to case positive rates only). Of note, the modelling framework characterises the uneven susceptibility landscape of the population to infection/re-infection arising from varying vaccination as well as past infection histories by known SARS-CoV-2 variants. In turn, the authors were able to compute a risk factor associated with different SARS-CoV-2 variants depending on the proportions of the population that were vaccinated. To validate their model, the authors initialised their simulations with empirical parameters similar to those observed in Austria during the initial introduction of the Delta variant in June 2021 and fitted their model outputs to observed outbreak incidence rates. Additionally, the authors also performed simulations of an outbreak introduced by a hypothetical novel immune escape variant. The main conclusion from both simulations is that a proactive control is more effective in control as opposed to reacting to case positivity rates alone.

However, the case positivity rate and Ne computed by the model is the true underlying value assuming 100% detection rate. This is never the case as different countries have different testing capacities and strategies of varying effectiveness (e.g. is contact tracing performed to detect asymptomatic as well or just by symptomatic testing? Even if there is contact tracing, it will never perfectly capture all transmission chains). Given that the controller reacts to these statistics computed from reported case counts, the effects of case reporting rates should be accounted for.

More critically, the key finding of this work that a proactive approach to NPI is more effective fell short in novelty, especially since the authors gave such an elaborate treatment to the differential immune landscape of the population. Previous studies have already shown that earlier, proactive implementation of NPIs always work better in lowering transmissions without explicitly modelling for differential immunity landscape (e.g. see https://www.science.org/doi/10.1126/scitranslmed.abg4262, https://www.thelancet.com/journals/laninf/article/PIIS1473-3099(20)30162-6/fulltext). We would not expect that a new wave of transmissions due to an immune escape variant would necessarily render earlier implementations of NPIs to have less relative impact. There are more meaningful hypothetical questions that can be queried by the model. For instance, what is the extent and duration that NPIs are needed to control future waves for countries with different vaccination coverage? Furthermore, what differences in future outbreaks would we expect for countries such as Singapore and New Zealand, which have previously kept local transmissions low (minimal infection-induced immunity) and had/expected to have high vaccination coverage, compared to others such as the U.K. that had experienced higher infection numbers (more heterogeneous immunity landscape from both infection and vaccination) in the past? Will their need and the strength of which NPIs be applied in these future outbreaks be similar?

6. PLOS authors have the option to publish the peer review history of their article (what does this mean?). If published, this will include your full peer review and any attached files.

**Do you want your identity to be public for this peer review?** For information about this choice, including consent withdrawal, please see our Privacy Policy.

Reviewer #1: **Yes: **Anna Bershteyn

Reviewer #2: No

---

## [Decision Letter · Decision Letter 1]

31 Jan 2022

PGPH-D-21-00695R1

Robust models of SARS-CoV-2 heterogeneity and control

Dear Dr. Johnson,

Thank you for submitting your manuscript to PLOS Global Public Health. After careful consideration, we feel that it has merit but does not fully meet PLOS Global Public Health’s publication criteria as it currently stands. Therefore, we invite you to submit a revised version of the manuscript that addresses the points raised during the review process.

We look forward to receiving your revised manuscript.

Kind regards,

Brooke E. Nichols

Academic Editor

Journal Requirements:

Additional Editor Comments (if provided):

In addition to the reviewer's final minor comments:

page 5-14 are functionally unreadable for the average 'global public health' reader. The information is essential for the manuscript, but should be summarised in just a few paragraphs- and the text that exists in the methods now should be moved to supplementary material. As an example to how complicated mathematical models can be communicated to global health audience, take a look at this article (https://journals.plos.org/plosmedicine/article?id=10.1371/journal.pmed.1002152). That manuscript includes the right amount of detail in the methods section, and has moved much of the mathematical language to the appendix.

Reviewers' comments:

Reviewer's Responses to Questions

**Comments to the Author**

1. If the authors have adequately addressed your comments raised in a previous round of review and you feel that this manuscript is now acceptable for publication, you may indicate that here to bypass the “Comments to the Author” section, enter your conflict of interest statement in the “Confidential to Editor” section, and submit your "Accept" recommendation.

Reviewer #1: (No Response)

Reviewer #2: All comments have been addressed

2. Does this manuscript meet PLOS Global Public Health’s publication criteria? Is the manuscript technically sound, and do the data support the conclusions? The manuscript must describe methodologically and ethically rigorous research with conclusions that are appropriately drawn based on the data presented.

Reviewer #1: Yes

Reviewer #2: Yes

3. Has the statistical analysis been performed appropriately and rigorously?

Reviewer #1: Yes

Reviewer #2: Yes

4. Have the authors made all data underlying the findings in their manuscript fully available (please refer to the Data Availability Statement at the start of the manuscript PDF file)?

Reviewer #1: Yes

Reviewer #2: Yes

5. Is the manuscript presented in an intelligible fashion and written in standard English?

Reviewer #1: Yes

Reviewer #2: Yes

6. Review Comments to the Author

Reviewer #1: Thanks to the authors for very nice progress on improving this manuscripts. The authors have addressed many of this Reviewer's comments, but have misunderstood the first comment about estimating Re in practice. For this, put yourself in the shoes of a health department. You will not know the parameters of Equation 11 with certainty. For example, with each successive pandemic wave there is tremendous uncertainty about R0, IFR, immune evasion, vaccine evasion, and other basic properties of the disease. That is why it is not typical for decision-makers to use even the proposed "default" controller. What you do know if the early trend in detected cases -- if it is growing, and if so, how fast -- but this is subject to time-varying case ascertainment bias. What "default" control rules can decision-makers use in practice, subject to these uncertainties?

The addition of quantitative results comparing the two controllers is helpful, but they are scattered throughout the text. Is it possible to systematize the quantitative results comparison in a table, emphasizing estimates that are of greatest public health interest (differences in cumulative infections and deaths, peak hospital occupancy, etc.)?

Lastly, I encourage the authors to drop the use of "Omega" to refer to the Omicron variant, since this term refers to a different SARS-CoV-2 variant with rather different properties.

Reviewer #2: The authors have adequately and rigorously addressed my previous comments. Wonderful piece of work, congratulations.

7. PLOS authors have the option to publish the peer review history of their article (what does this mean?). If published, this will include your full peer review and any attached files.

**Do you want your identity to be public for this peer review?** For information about this choice, including consent withdrawal, please see our Privacy Policy.

Reviewer #1: No

Reviewer #2: No

---

## [Editor Report · Decision Letter 2]

6 Apr 2022

Robust models of disease heterogeneity and control, with application to the SARS-CoV-2 epidemic

PGPH-D-21-00695R2

Dear Dr. Johnson,

We are pleased to inform you that your manuscript 'Robust models of disease heterogeneity and control, with application to the SARS-CoV-2 epidemic' has been provisionally accepted for publication in PLOS Global Public Health.

Best regards,

Brooke E. Nichols

Academic Editor